# Efficient Denoising Diffusion via Probabilistic Masking

## Abstract

Diffusion models have exhibited remarkable advancements in generating high-quality data. However, a critical drawback of these models is their computationally intensive inference process, which requires a large number of timesteps to generate a single sample. Existing methods address this challenge by decoupling the forward and reverse processes, and they rely on handcrafted rules (e.g., uniform skipping) for sampling acceleration, leading to the risk of discarding important steps and deviating from the optimal trajectory. In this paper, we propose an Efficient Denoising Diffusion method via Probabilistic Masking (EDDPM) that can identify and skip the redundant steps during training. To determine whether a timestep should be skipped or not, we employ probabilistic reparameterization to continualize the binary determination mask. The mask distribution parameters are learned jointly with the diffusion model weights. By incorporating a real-time sparse constraint, our method can effectively identify and eliminate unnecessary steps during the training iterations, thereby improving inference efficiency. Notably, as the model becomes fully trained, the random masks converge to a sparse and deterministic one, retaining only a small number of essential steps. Empirical results demonstrate the superiority of our proposed EDDPM over the state-of-the-art sampling acceleration methods across various domains. EDDPM can generate high-quality samples with only 20% of the steps for time series imputation and achieve 4.89 FID with 5 steps for CIFAR-10. Moreover, when starting from a pretrained model, our method efficiently identifies the most informative timesteps within a single epoch, which demonstrates the potential of EDDPM to be a practical tool to explore large diffusion models with limited resources.

## 1 Introduction

Diffusion models have emerged as a powerful generative technique, achieving unprecedented success in various domains, including image generation (Ho et al., 2020; Saharia et al., 2022; Dhariwal & Nichol, 2021), speech synthesis (Kong et al., 2020; Jeong et al., 2021), text generation (Hoogeboom et al., 2021; Li et al., 2022), 3D shape generation (Luo & Hu, 2021) and time series forecasting and imputation (Tashiro et al., 2021; Rasul et al., 2021). These models employ an iterative sampling procedure to generate each sample by progressively removing noise from random initial vectors.

One significant drawback of diffusion models is their reliance on a large number of denoising steps, ranging from hundreds to thousands, to transform Gaussian noise into a sample. As a result, diffusion models are considerably slower compared to other generative models like GANs (Brock et al., 2018). In recent years, several acceleration techniques have been proposed to address this issue, which can be divided into learning-free and learning-based methods according whether additional training is required. It is worth noting that learning-free methods (Song et al., 2020a; Bao et al., 2021; Liu et al., 2021; Bao et al., 2022) often employ handcrafted rules, whereas learning-based methods (Watson et al., 2021a;b; Dockhorn et al., 2022; Salimans & Ho, 2021; Luhman & Luhman, 2021) decouple the training and inference schedules. This decoupling allows for separate learning of the training and sampling schedules. However, both approaches have the potential to result in suboptimal performance. Therefore, exploring the determination of the optimal sampling step during training is a promising direction worth investigating.

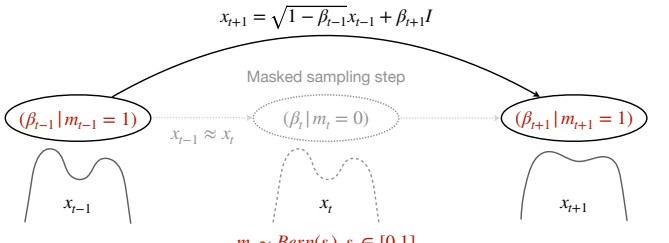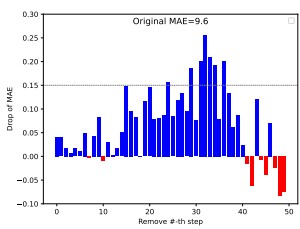

Figure 1: The left sub-figure shows the parametic probabilistic masking method, in which the masks are determined by the Bernoulli distribution. The steps with 0-valued masks will be skipped. The right sub-figure shows the performance changes when we simply remove a single step. The red rectangles indicate that the sample quality can be improved after removing that step. The results are collected from CSDI (Tashiro et al., 2021) model with the Air-quality dataset. It shows that for most steps, performance drop of removing them is negligible, i.e., lower than 0.15. It can be expected that more promising results can be achieved when more advanced approaches are employed.

In this paper, we propose an **e**fficient **d**enoising **d**iffusion **m**odel (EDDPM) to enhance the sampling efficiency. The fundamental concept is illustrated in Figure 1. The sub-figure on the right highlights redundant sampling steps in the denoising process. Removing these steps has a negligible effect on the quality of the samples or may even improve the sample quality. To automatically and safely skip redundant steps in the forward and reverse diffusion processes, we propose EDDPM, which is a diffusion model equipped with a novel probabilistic masking module. This module gradually identifies and masks the less informative steps. To be precise, we assign a binary probabilistic mask $m_i$ to each diffusion step $i$, indicating whether it should be skipped ($m_i = 0$) or kept ($m_i = 1$). Since searching for the globally optimal steps for diffusion models is an intractable discrete optimization problem, we address it by continualizing it through probabilistic reparameterization. We parameterize $m_i$ to be a Bernoulli random variable with probability $s_i$ set to $1$ and probability $1 - s_i$ set to 0. Consequently, we can use the sum of $s_i$ to control the model efficiency, which can finally be encoded into a sparse constraint. Therefore, the training problem is continualized into optimizing the denoising diffusion model under the sparsity constraint.

By jointly training the denoising diffusion model and optimizing the mask distribution parameters, we are able to automatically identify and eliminate redundant steps. Our method possesses an appealing feature: as a result of the sparse constraint applied to the distribution parameters, the majority of probabilities $s_i$ will converge to either 0 or 1 upon full training. Consequently, the masks tend to converge to nearly deterministic ones after training and the redundant steps can thus be safely removed.We conducted extensive quantitative and qualitative evaluations on image synthesis and time series imputation tasks to validate the effectiveness of our method. For instance, in time series imputation, our method achieves significant performance improvements, generating high-quality samples using only 20% of the original steps. Moreover, it achieves an impressive FID score of 4.89 on CIFAR-10 with just 5 steps. Another advantage of our method is that it only requires one epoch of fine-tuning on a pretrained model to determine the most informative denoising steps. This makes it feasible to explore large diffusion models even with limited resources, which is particularly valuable for the research in academia. The main contributions of this work can be summarized as follows:

- We propose an efficient denoising diffusion model via probabilistic masking, offering the following advantages:
    - Our method can identify and remove redundant denoising steps during training, eliminating the need for handcrafted skipping rules.
    - Most of our probabilistic masks converge to deterministic values after full training, allowing for the safe removal of redundant steps.
    - The training efficiency in the later stages is significantly improved. As the training process goes on, most $s_i$'s would get closer to either 0 or 1. Thus, our EDDPM would automatically select the uninformative steps (i.e., $s_i$ is small) with low probability and always focus on the informative ones (i.e., $s_i$ is close to 1) to train the model weights.
- We verify the proposed EDDPM method on two domain tasks. In the time series imputation benchmarks, the empirical results on Healthcare and Air-quality datasets demonstrate that

EDDPM outperforms the state-of-the-art sampling acceleration methods. It achieves comparable or even superior performance compared to the original baselines with only 20% denosing steps. Moreover, EDDPM achieves a 4.89 FID with only 5 steps for CIFAR-10.

- EDDPM demonstrates impressive performance in diffusion model compression. Starting from a pretrained diffusion model, our method can compress it within a single epoch of fine-tuning. This highlights the potential of EDDPM as a practical tool for exploring large diffusion models with limited resources.

## 2 RELATED WORKS

In this section, we first review the development and applications of diffusion models, Then, we introduce the recent work on improving the sampling efficiency of diffusion models.

**Development and applications of DPMs.** Diffusion probabilistic models (DPMs) are firstly introduced by (Sohl-Dickstein et al., 2015) that they can convert one distribution into a target distribution, in which each diffusion step is tractable. Bordes et al. (2016) improved DPMs by a infusion training procedure that requires slightly shorter generation trajectory. Ho et al. (2020) proposed a new diffusion model, called Denoising Diffusion Model (DDPM), and a weighted variational bound objective by connecting the DPMs and denoising score matching methods (Song & Ermon, 2019). Song et al. (2020a) generalized the DDPMs to non-Markovian diffusion processes which lead to "short" generative Markov chains that can increase sample efficiency. With above the important improvements, DPMs show great potential in various applications, including speech synthesis (Kong et al., 2020; Jeong et al., 2021), 3D shape generation (Luo & Hu, 2021), image super-resolution (Saharia et al., 2022), text generation (Hoogeboom et al., 2021; Li et al., 2022) and probabilistic time series forecasting(Rasul et al., 2021) and imputation (Tashiro et al., 2021) Previous studies have shown deep learning models can capture the temporal dependency of time series and give more accurate imputation than statistical methods (Fortuin et al., 2020; Mulyadi et al., 2021; Bonilla & Chai, 2007). Rasul et al. (2021) used DPMs for multivariate probabilistic time series forecasting and achieved the state-of-the-art performance. CSDI (Tashiro et al., 2021) is a conditional score-based diffusion model and it is used to generate the missing values in the time series. To our knowledge, prior works have not explore the acceleration of DPMs on time series domain.

**Acceleration of DPMs.** Following the survey (Yang et al., 2022), we divide the existing efficient sampling methods into two categories, i.e., learning-free and learning-based on methods based on whether they require an additional learning process after the diffusion model has been trained. The learning-free approaches accelerate the sampling process by discretizing either the reverse-time stochastic differential equations (SDE) (Dockhorn et al., 2021; Song et al., 2020b; Jolicoeur-Martineau et al., 2020) or the probability flow ordinary differential equations (ODE) (Liu et al., 2021; Song et al., 2020a; Zhang et al., 2022; Lu et al., 2022). We notice that these methods always use handcrafted steps to select the denosing steps. As our proposed method belongs to the learning based category, here we mainly review the recent studies on learning-based efficient sampling methods (Watson et al., 2021a;b; Dockhorn et al., 2022; Salimans & Ho, 2021; Luhman & Luhman, 2021), which find efficient denoising trajectories by optimizing some objective or using knowledge distillation.For example, Watson et al. (2021b) used the dynamic programming algorithm to search the informative diffusion steps. Xiao et al. (2021) compressed the diffusion process by combining the GANs and DPMs, the efficiency is improved since larger step size is allowed. San-Roman et al. (2021) estimated the level of noise by training a separate model, and modified the denoising process dynamically to match the predicted noise level. Dockhorn et al. (2022) derive a second-order solver for accelerating synthesis by training a additional head on top of the first-order score network. Knowledge distillation is adopted in (Salimans & Ho, 2021; Luhman & Luhman, 2021) to distill the full sampling process into a faster sampler.

Through promising results are reported in existing studies, it is worth noting that learning-free methods often employ handcrafted rules, whereas learning-based methods usually decouple the training and inference schedules. However, both approaches have the potential to result in suboptimal performance. To overcome the separation of the training and the inference processes, we propose to progressively remove the redundant diffusion steps in the training through a probabilistic parameterization approach.

**Notations:** Let $\| \cdot \|_1$ be the $\ell_1$ norm of a vector. We denote $\mathbf{1} \in \mathbb{R}^n$ and $\mathbf{0} \in \mathbb{R}^n$ to be the vectors with all components equal to 1 and 0. In addition, $\{0,1\}^n$ is a set of $n$-dimensional vectors with each coordinate valued in $\{0,1\}$.

## 3 BASICS

In this section, for the convenience of presenting our method EDDPM in Section 4, we introduce the basics of denoising diffusion probabilistic models.

Starting from a sample $\mathbf{x}_0$, a *diffusion process* or *forward process* is defined as a $T$-step Markov chain, where Gaussian noise is gradually injected into $\mathbf{x}_0$. That is

$$q(\mathbf{x}_{1:T}|\mathbf{x}_0) = \prod_{t=1}^{T} q(\mathbf{x}_t|\mathbf{x}_{t-1}) \quad \text{with} \quad q(\mathbf{x}_t|\mathbf{x}_{t-1}) = \mathcal{N}(\mathbf{x}_t|\sqrt{1-\beta_t}\mathbf{x}_{t-1}, \beta_t\mathbf{I}), \tag{1}$$

where the scalar parameter $\beta_t$ $(t = 1, \ldots, T)$ determines the variance of the noise added at each diffusion step, subject to $0 < \beta_t < 1$. $\mathbf{x}_1, ..., \mathbf{x}_T$ are latent variables in the same space as $\mathbf{x}_0$. It can be verified that the diffusion schedule in Eqn.(1) can guarantee $x_T$ would be close to a standard Gaussian noise, i.e., $\mathcal{N}(\mathbf{x}_T; \mathbf{0}, \mathbf{I})$, when $T$ is sufficiently large. Notice that $\mathbf{x}_t$ at an arbitrary timestep $t$ takes the form of:

$$q(\mathbf{x}_t|\mathbf{x}_0) = \mathcal{N}(\mathbf{x}_t; \sqrt{\alpha_t}\mathbf{x}_0, (1-\alpha_t)\mathbf{I}) \tag{2}$$

where $\alpha_t = \prod_{s=1}^{t} \hat{\alpha}_s$, $\hat{\alpha}_t = 1 - \beta_t$. This property enables us to sample $\mathbf{x}_t$ at any timestep $t$ in training without going through $x_i, i \leq t$ one by one.

The *reverse process* is modelled as another Markov chain parameterized by $\theta$. To be precise, it starts from $p(\mathbf{x}_T) = \mathcal{N}(\mathbf{x}_T; \mathbf{0}, \mathbf{I})$ and

$$p_\theta(\mathbf{x}_{0:T}) = p(\mathbf{x}_T) \prod_{t=1}^{T} p_\theta(\mathbf{x}_{t-1}|\mathbf{x}_t) \quad \text{with} \quad p_\theta(\mathbf{x_{t-1}}|\mathbf{x_t}) = \mathcal{N}(\mathbf{x}_{t-1}; \mu_\theta(\mathbf{x}_t, t), \sigma_t^2\boldsymbol{I}). \tag{3}$$

The parameters $\theta$ of the reverse process can be learned by maximizing the following variational lower bound on the training set, i.e.,

$$\mathbb{E}_{q(\mathbf{x}_0)} \log p_\theta(\mathbf{x}_0) \geq \mathbb{E}_{q(\mathbf{x}_0, \mathbf{x}_1, \ldots, \mathbf{x}_T)} \log \frac{p_\theta(\mathbf{x}_{0:T})}{q(\mathbf{x}_{1:T}|\mathbf{x}_0)}. \tag{4}$$

Minimizing the objective function in Eqn.(4) is equivalent to minimizing the distance between $p_\theta(\mathbf{x_{t-1}}|\mathbf{x_t})$ against forward process posteriors $q(\mathbf{x}_{t-1}|\mathbf{x}_t, \mathbf{x}_0)$, which is actually a Gaussian distribution, i.e.,

$$q(\mathbf{x}_{t-1}|\mathbf{x}_t, \mathbf{x}_0) = \mathcal{N}(\mathbf{x}_{t-1}; \tilde{\mu}_t(\mathbf{x}_t, \mathbf{x}_0), \tilde{\beta}_t\mathbf{I}), \tag{5}$$

$$\text{where } \tilde{\mu}_t(\mathbf{x}_t, \mathbf{x}_0) := \frac{\sqrt{\alpha_{t-1}}\beta_t}{1-\alpha_t}\mathbf{x}_0 + \frac{\sqrt{\hat{\alpha}_t}(1-\alpha_{t-1})}{1-\alpha_t}\mathbf{x}_t, \quad \tilde{\beta}_t := \frac{1-\alpha_{t-1}}{1-\alpha_t}\beta_t.$$

By parameterizing $p_\theta(\mathbf{x}_{t-1}|\mathbf{x}_t)$ as $\mathcal{N}(\mathbf{x}_{t-1}, \mu_\theta(\mathbf{x}_t, t), \sigma_t^2\mathbf{I})$ with

$$\mu_\theta(\mathbf{x}_t, t) = \tilde{\mu}_t\left(\mathbf{x}_t, \frac{1}{\sqrt{\alpha_t}}(\mathbf{x}_t - \sqrt{1-\alpha_t}\epsilon_\theta(\mathbf{x}_t, t))\right),$$

and letting $\epsilon \sim \mathcal{N}(\mathbf{0}, \mathbf{I})$, the overall training problem of diffusion model can be written as

$$\min_\theta \mathbb{E}_{\mathbf{x}_0, \epsilon, t}\left[\frac{\beta_t^2}{2\sigma_t^2\hat{\alpha}_t(1-\alpha_t)}\left\|\epsilon - \epsilon_\theta\left(\sqrt{\alpha_t}\mathbf{x}_0 + \sqrt{1-\alpha_t}\epsilon, t\right)\right\|^2\right]. \tag{6}$$

## 4 METHOD

In this section, we present our efficient denoising diffusion probabilistic model EDDPM by first introducing our probabilistic masking approach and then presenting the detailed training procedure. The detailed derivation procedure for our method is given in the appendix.

### 4.1 PROBABILISTIC MASKING FOR DIFFUSION MODELS

As shown in Figure 1, our basic idea is to assign a binary mask $m_t$ to determine whether this time step $t$ should be skipped (i.e., $m_t = 0$) or not (i.e., $m_t = 1$) and then jointly learn these masks with the diffusion model parameters.

In EDPPM, we multiply each variance $\beta_t$ with the binary mask $m_t$. Therefore, $m_t = 0$ means the diffusion step $t$ can be skipped, since the injected noise in Eqn.(1) would be $\mathbf{0}$. Thus, the $\ell_1$-norm of $\mathbf{m}$, i.e., $\|\mathbf{m}\|_1$, can be used to control the number of steps the diffusion model goes through. Notice that, in this way, $\mathbf{x}_t$ is still a Gaussian random variable. To be precise,

$$\mathbf{x}_t \sim \mathcal{N}(\sqrt{\alpha_t(\mathbf{m})}\mathbf{x}_0, (1 - \alpha_t(\mathbf{m}))\mathbf{I}) \text{ where } \alpha_t(\mathbf{m}) = \prod_{i=1}^t \hat{\alpha}_i(\mathbf{m}), \text{ and } \hat{\alpha}_t(\mathbf{m}) = 1 - \beta_t m_t. \quad (7)$$

It implies that we can sample $\mathbf{x}_t$ at any time step $t$ without going through the former steps $0$ to $t-1$. The variational low bound for diffusion models can be written as:

$$\mathcal{L}_\theta(\mathbf{x}_0, \epsilon, \mathbf{m}) = -\mathbb{E}_q \left[ \log p(\mathbf{x}_T | \mathbf{m}) + \sum_{t \geq 1} \log \frac{p_\theta(\mathbf{x}_{t-1} | \mathbf{x}_t, \mathbf{m})}{q(\mathbf{x}_t | \mathbf{x}_{t-1}, \mathbf{m})} \right], \quad (8)$$

where the masked reverse process corresponding to Eqn. (3) becomes

$$p_\theta(\mathbf{x}_{t-1} | \mathbf{x}_t, \mathbf{m}) = \mathcal{N}(\mathbf{x}_{t-1}; \boldsymbol{\mu}_\theta(\mathbf{x}_t, \mathbf{m}, t), \sigma_t^2(\mathbf{m})\mathbf{I}), \quad (9)$$

and the masked forward process posterior takes the form of

$$q(\mathbf{x}_{t-1} | \mathbf{x}_t, \mathbf{x}_0, \mathbf{m}) = \mathcal{N}(\mathbf{x}_{t-1}; \tilde{\mathbf{u}}(\mathbf{x}_t, \mathbf{x}_0, \mathbf{m}), \tilde{\beta}_t(\mathbf{m})\mathbf{I}), \text{ with}$$

$$\tilde{\mathbf{u}}_t(\mathbf{x}_t, \mathbf{x}_0, \mathbf{m}) = \frac{\sqrt{\alpha_{t-1}(\mathbf{m})}\beta_t m_t}{1 - \alpha_t(\mathbf{m})}\mathbf{x}_0 + \frac{\sqrt{\hat{\alpha}_t(\mathbf{m})}(1 - \alpha_{t-1}(\mathbf{m}))}{1 - \alpha_t(\mathbf{m})}\mathbf{x}_t, \tilde{\beta}_t(\mathbf{m}) = \frac{1 - \bar{\alpha}_{t-1}(\mathbf{m})}{1 - \bar{\alpha}_t(\mathbf{m})}m_t\beta_t.$$

The problem of training a sparse diffusion can be formulated naturally into

$$\min_\theta \mathbb{E}_{\mathbf{x}_0, \epsilon, t} \left[ C_t \left\| \epsilon - \epsilon_\theta \left( \sqrt{\alpha_t(\mathbf{m})}\mathbf{x}_0 + \sqrt{1 - \alpha_t(\mathbf{m})}\epsilon, t \right) \right\|^2 \right], s.t. \quad \|\mathbf{m}\|_1 \leq K, \mathbf{m} \in \{0, 1\}^T,$$

where $C_t = \frac{\beta_t^2 m_t}{2\sigma_t^2 \hat{\alpha}_t(\mathbf{m})(1 - \alpha_t(\mathbf{m}))}$ with $K$ is a positive integer controls the process complexity, $T$ is the total length of the diffusion process.

Notice that the above formulation involves a discrete optimization problem, which is hard to solve and thus cannot be applied in practice. To address this issue, we contiualize the training problem by probabilistic masking. That is, we reparameterize $\mathbf{m}$ into a binary random vector with each component $m_t$ being an independent Bernoulli random variable with probability $\mathbf{s}_t \in [0, 1]$ to be 1 and $1 - s_t$ to be 0. Then, we can relax the above discrete optimization problem into the following continuous one:

$$\min_{\theta, \mathbf{s}} \Phi(\theta, \mathbf{s}) := \mathbb{E}_{\mathbf{m} \sim p(\mathbf{m}|\mathbf{s})} \mathbb{E}_{\mathbf{x}_0, \epsilon, t | \mathbf{m}} \mathcal{L}_\theta^t(\mathbf{x}_0, \epsilon, \mathbf{m}), \quad s.t. \quad \|\mathbf{s}\|_1 \leq K, \mathbf{s} \in [0, 1]^T, \quad \text{(EDDPM)}$$

$$\text{where } \mathcal{L}_\theta^t(\mathbf{x}_0, \epsilon, \mathbf{m}) = C_t \left\| \epsilon - \epsilon_\theta \left( \sqrt{\alpha_t(\mathbf{m})}\mathbf{x}_0 + \sqrt{1 - \alpha_t(\mathbf{m})}\epsilon, t \right) \right\|^2.$$

Notice that $C_t = 0$ when $m_t = 0$, therefore, we do not need to update the model in this case.
**Discussion.** Nice properties of our EDDPM are summarized as follows:

- Given the mask $\mathbf{m}$, since $\mathbf{x}_t$ are Gaussian random variables, we can sample them at any time step $t$ without going through the former diffusion steps. This would enable us to train the model efficiently.
- Due to the constraints on $\mathbf{s}$, i.e., $\|\mathbf{s}\|_1 \leq K$ and $\mathbf{s} \in [0, 1]^T$, the optimal $\mathbf{s}$ would be sparse and most of its components would be either 0 or 1. See Section C in the appendix for details. This would lead us to the three advantages below:
  - When fully trained, the mask $\mathbf{m}$ sampled from Bernoulli distribution $p(\mathbf{m}|\mathbf{s})$ would become sparse. Thus, the length of the denoising steps can be significantly reduced and inference efficiency can be improved.

- Since the mask $\mathbf{m}$ would be nearly deterministic after training, the steps with $0$ valued masks can be safely discarded. Therefore, the undesired randomness in sampling the final diffusion model is eliminated.

- The training efficiency in the late stage can be improved. The reason is that as the training process goes on , most $\mathbf{s}_i$'s would get closer to either 0 or 1. Therefore our EDDPM would automatically select the uninformative steps (i.e., $\mathbf{s}_i$ is small) with low probability and always focus on the informative ones (i.e., $\mathbf{s}_i$ is close to 1) to train the model weights. Thus, the training efficiency can be improved, which is verified in the experimental results (see Figure 2).

### 4.2 UPDATING MASKING SCORES

We adopt stochastic optimization algorithms to train our model EDDPM. Therefore, the key technique is to estimate the stochastic gradient. Notice that

$$\nabla_\theta \Phi(\theta, \mathbf{s}) = \mathbb{E}_{\mathbf{m} \sim p(\mathbf{m}|\mathbf{s})} \mathbb{E}_{\mathbf{x}_0, \epsilon, t|\mathbf{m}} \nabla_\theta \mathcal{L}_\theta^t(\mathbf{x}_0, \epsilon, \mathbf{m}). \tag{10}$$

Hence, $\nabla_\theta \mathcal{L}_\theta^t(\mathbf{x}_0, \epsilon, \mathbf{m})$ is an unbiased estimation of $\nabla_\theta \Phi(\theta, \mathbf{s})$. Below, we introduce the estimation of $\nabla_\mathbf{s} \Phi(\theta, \mathbf{s})$.

**[Gradient Computation $\nabla_\mathbf{s} \Phi(\theta, \mathbf{s})$]** We adopt the policy gradient method to estimate the gradient. To be precise,

$$\begin{aligned}
\nabla_\mathbf{s} \Phi(\theta, \mathbf{s}) &= \nabla_\mathbf{s} \sum_\mathbf{m} \left[ \mathbb{E}_{\mathbf{x}_0, \epsilon, t|\mathbf{m}} \mathcal{L}_\theta^t(\mathbf{x}_0, \epsilon, \mathbf{m}) \right] p(\mathbf{m}|\mathbf{s}) \\
&= \sum_\mathbf{m} \left[ \mathbb{E}_{\mathbf{x}_0, \epsilon, t|\mathbf{m}} \mathcal{L}_\theta^t(\mathbf{x}_0, \epsilon, \mathbf{m}) \right] \nabla_\mathbf{s} p(\mathbf{m}|\mathbf{s}) \\
&= \sum_\mathbf{m} \left[ \mathbb{E}_{\mathbf{x}_0, \epsilon, t|\mathbf{m}} \mathcal{L}_\theta^t(\mathbf{x}_0, \epsilon, \mathbf{m}) \nabla_\mathbf{s} \ln p(\mathbf{m}|\mathbf{s}) \right] p(\mathbf{m}|\mathbf{s}) \\
&= \mathbb{E}_{\mathbf{m} \sim p(\mathbf{m}|\mathbf{s})} \mathbb{E}_{\mathbf{x}_0, \epsilon, t|\mathbf{m}} \mathcal{L}_\theta^t(\mathbf{x}_0, \epsilon, \mathbf{m}) \nabla_\mathbf{s} \ln p(\mathbf{m}|\mathbf{s}). \tag{11}
\end{aligned}$$

Therefore, $\mathcal{L}_\theta^t(\mathbf{x}_0, \epsilon, \mathbf{m}) \nabla_\mathbf{s} \ln p(\mathbf{m}|\mathbf{s})$ is a stochastic gradient of $\Phi(\theta, \mathbf{s})$.

Based on Eqn.(10) and (11), we know that during training, we can estimate the gradients $\nabla_\theta \Phi(\theta, \mathbf{s})$ and $\nabla_\mathbf{s} \Phi(\theta, \mathbf{s})$ by sampling a random mask $\mathbf{m}$ and a Gaussian noise $\epsilon$.

**[Gradually Increasing Masking Rate]** To control the model complexity, we denote the final masking rate as $\gamma_f$, that is

$$K = \gamma_f T.$$

Then, to stabilize the training process, we increase the masking rate gradually to make a smooth transformation from a full diffusion process to a sparse process. We utilize the increase function of Zhu & Gupta (2017):

$$\gamma_e = \begin{cases} 1, & \text{if } e < e_1, \\ \gamma_f + (1 - \gamma_f) \left( 1 - \frac{e - e_1}{N - e_1} \right)^3, & \text{otherwise,} \end{cases} \tag{12}$$

where $N$ is the training epoch, $\gamma_e$ is the ratio of the remaining steps in the current epoch $e$. $e_1$ is a positive integer indicating that we train the entire denosing steps in the first $e_1$ epochs.

After obtaining the gradients, $\theta$ and $\mathbf{s}$ are updated by projected gradient descent as follows:

$$\theta = \theta - \eta \nabla_\theta \Phi(\theta, \mathbf{s}) \text{ and } \mathbf{s} = \text{proj}_\mathcal{S} \left( \mathbf{s} - \eta \nabla_\mathbf{s} \Phi(\theta, \mathbf{s}) \right), \tag{13}$$

where $\mathcal{S} = \{ \mathbf{s} \in \mathbb{R}^T : ||\mathbf{s}||_1 \leq K_e, \mathbf{s} \in [0, 1]^T \}$ with $K_e = \gamma_e T$. The projection can be efficiently computed with the details given in Theorem 1 of the appendix.

Our training method can be integrated with general stochastic optimization algorithms flexibly. The detailed steps of our algorithm are given in the appendix.

Table 1: CIFAR-10 image generation measured in FID. "-" means this result is not provided in the corresponding paper. The underlined result is the second best method. NCSN++ *w/* TDAS achieves lower FID than our method as it adopts different base model.

| Method | Denoising steps | | | | | | |
|---|---|---|---|---|---|---|---|
| | 5 | 10 | 25 | 50 | 100 | 200 | 1000 |
| Learning-free methods | | | | | | | |
| DDPM (Ho et al., 2020) | - | 233.41 | 125.05 | 66.28 | 31.36 | 12.96 | 3.04 |
| DDIM (Song et al., 2020a) | 41.6 | 21.31 | 10.70 | 7.74 | 6.08 | 5.07 | 4.13 |
| SN-DDPM (Bao et al., 2022) | - | 24.06 | 6.91 | 4.63 | 3.67 | 3.31 | 3.65 |
| SN-DDIM (Bao et al., 2022) | - | 12.19 | 4.28 | 3.39 | 3.23 | 3.22 | 3.65 |
| NPR-DDPM (Bao et al., 2022) | - | 32.35 | 10.55 | 6.18 | 4.52 | 3.57 | 4.10 |
| NPR-DDIM (Bao et al., 2022) | - | 13.34 | 5.38 | 3.95 | 3.53 | 3.42 | 3.72 |
| Analytic-DDPM (Bao et al., 2021) | - | 34.26 | 11.60 | 7.25 | 5.40 | 4.01 | 4.03 |
| Analytic-DDIM (Bao et al., 2021) | - | 14.00 | 5.81 | 4.04 | 3.55 | 3.39 | 3.74 |
| S-PNDM (Liu et al., 2021) | 35.9 | 11.6 | - | 5.18 | 4.34 | - | 3.80 |
| F-PNDM (Liu et al., 2021) | - | 7.03 | - | 3.95 | 3.72 | - | 3.70 |
| Learning-based methods | | | | | | | |
| GGDM (Watson et al., 2021a) | 13.77 | 8.23 | 4.25 | - | - | - | - |
| DPM-solver (Lu et al., 2022) | - | 4.70 | - | - | - | - | - |
| GENIE (Dockhorn et al., 2022) | 11.2 | 5.28 | 3.64 | - | - | - | - |
| NCSN *w/* TDAS (Ma et al., 2022) | - | - | - | - | - | 72.92 | 23.56 |
| NCSN++ *w/* TDAS (Ma et al., 2022) | - | - | - | - | 7.78 | **2.97** | - |
| EDDPM | **4.89** | **4.34** | **3.59** | **3.34** | **3.21** | 3.19 | **3.03** |

## 5 EXPERIMENT

In this section, we evaluate the the effectiveness of EDDPM on two applications of DDPM, i.e., image synthesis and multivariate time series imputation. We follow the experimental settings in the existing studies (Ho et al., 2020; Song et al., 2020a; Watson et al., 2021a) to ensure a fair comparison.

**Datasets.** We use the CIFAR-10 dataset (Krizhevsky et al., 2009) (50k images of resolution $32 \times 32$) for image synthesis, and Healthcare (Silva et al., 2012) and Air-quality (Tashiro et al., 2021) for the time series imputation experiments.

**Baselines.** The following two different sets of baselines are used for time series imputation and image synthesis. For the time series task, we compared EDDPM to a variety of scheduling and acceleration techniques applicable to DDPMs: DDPM (Ho et al., 2020), DDIM (Song et al., 2020a), Analytic-DPM (Bao et al., 2021) and Extended-Analytic-DPM (Bao et al., 2022). Based on CSDI model (Tashiro et al., 2021), these methods are all re-implemented in our codebase. In addition to the above acceleration methods, we also compare PNDM (Liu et al., 2021), GGDM (Watson et al., 2021a), DPM-solver (Lu et al., 2022), GENIE (Dockhorn et al., 2022) and TDAS (Ma et al., 2022) on CIFAR-10 benchmark.

**Evaluation Metric.** In the time series task, we evaluate the performance on normalized data (zero mean and unit variance) by three commonly used metrics (Mean Absolute Error (MAE), Root Mean Squared Error (RMSE)) for probabilistic time series imputation. Following previous studies (Tashiro et al., 2021), we generate 100 samples to approximate the probability distribution over missing values and report the normalized average of CRPS for all missing values. The detailed formulations of these three metrics are provided in the appendix. In the image generation task, we use the Fréchet Inception Distance (FID) (Heusel et al., 2017) to evaluate generated 50K samples. The transformed feature is the 2048-dimensional vector output of the last layer of Inception-V3 (Szegedy et al., 2016).

### 5.1 MAIN RESULTS

In this section, we demonstrate the superior performance of our proposed EDDPM method in compressing diffusion steps for probabilistic time series imputation and image generation tasks, as compared to sampling acceleration methods. Additional visualization results can be found in the appendix.

As presented in Table 1, our proposed EDDPM method outperforms previous state-of-the-art sampling acceleration methods in terms of CIFAR-10 image generation, particularly at sampling steps of 5 and 10. These results illustrate that the decoupled forward and reverse processes fail to identify

Table 2: Comparising sampling acceleration methods in terms of **RMSE** results on variable denoising steps. $^{\dagger}$ indicate that the sampling is accelerated by quadratic skipping during inference, the others utilize uniform skipping. We highlight the best results that surpass the baseline (DDPM is trained with 100% sampling steps) in red color, which means our method generates high-quality time series with fewer denoising steps. The **bold** results show that our proposed EDDPM achieves better performance than other sampling acceleration methods.

| Dataset | Missing rate | Method | Denoising steps | | | | Baseline |
|---|---|---|---|---|---|---|---|
| | | | 10% | 25% | 40% | 50% | |
| Healthcare | 10% | DDPM$^{\dagger}$ | 0.934 | 0.774 | 0.682 | 0.646 | 0.549 |
| | | DDPM | 0.900 | 0.707 | 0.634 | 0.592 | |
| | | DDIM | 0.899 | 0.709 | 0.611 | 0.727 | |
| | | AnalyticDPM | 0.856 | 0.842 | 0.831 | 0.825 | |
| | | SN-DDPM | 1.094 | 1.112 | 1.128 | 1.146 | |
| | | NPR-DDIM | 0.818 | 0.804 | 0.820 | 0.844 | |
| | | Ours (EDDPM) | **0.582** | 0.545 | 0.529 | 0.505 | |
| | 50% | DDPM$^{\dagger}$ | 0.971 | 0.876 | 0.802 | 0.765 | 0.679 |
| | | DDPM | 0.961 | 0.815 | 0.747 | 0.706 | |
| | | DDIM | 0.962 | 0.821 | 0.746 | 0.755 | |
| | | AnalyticDPM | 0.976 | 0.902 | 0.896 | 0.865 | |
| | | SN-DDPM | 1.029 | 1.038 | 1.047 | 1.061 | |
| | | NPR-DDIM | 0.872 | 0.850 | 0.960 | 0.877 | |
| | | Ours (EDDPM) | **0.721** | 0.672 | 0.669 | 0.670 | |
| | 90% | DDPM$^{\dagger}$ | 0.990 | 0.963 | 0.931 | 0.911 | 0.823 |
| | | DDPM | 0.101 | 0.931 | 0.883 | 0.850 | |
| | | DDIM | 0.101 | 0.961 | 0.900 | 0.875 | |
| | | AnalyticDPM | 0.982 | 0.919 | 0.916 | 0.901 | |
| | | SN-DDPM | 1.179 | 1.115 | 1.105 | 1.095 | |
| | | NPR-DDIM | 0.953 | 0.913 | 0.914 | 0.917 | |
| | | Ours (EDDPM) | **0.851** | 0.811 | 0.809 | 0.815 | |
| Air-quality | 13% | | 4% | 10% | 20% | 40% | 19.212 |
| | | DDPM$^{\dagger}$ | 67.602 | 58.851 | 48.131 | 35.005 | |
| | | DDPM | 64.302 | 55.300 | 44.199 | 31.338 | |
| | | DDIM$^{\dagger}$ | 67.767 | 60.848 | 52.296 | 40.761 | |
| | | DDIM | 64.252 | 56.684 | 47.234 | 36.125 | |
| | | AnalyticDPM | 62.458 | 51.223 | 48.936 | 46.379 | |
| | | SN-DDPM | 78.885 | 73.111 | 69.656 | 69.430 | |
| | | SN-DDIM | 85.099 | 69.359 | 69.241 | 73.545 | |
| | | NPR-DDPM | 55.142 | 58.524 | 63.164 | 66.557 | |
| | | NPR-DDIM | 55.656 | 46.654 | 45.496 | 44.458 | |
| | | Ours (EDDPM) | **30.967** | **23.024** | 18.371 | 18.242 | |

the optimal sampling steps, while our EDDPM method can automatically identify more informative steps during training.

In Table 2, we evaluated the proposed EDDPM and alternative accelerated sampling methods. These methods utilize the same CSDI (Tashiro et al., 2021) backbone network for a pair-to-pair comparison. We have the following 3 observations. 1) The results demonstrate that, on Healthcare and Air-quality datasets, our proposed EDDPM can achieve better RMSE results than the baselines with 100% steps (blue text) even if $50\% \sim 80\%$ denoising steps are masked. 2) We also found that the denoising process with uniform skipping approach can obtain better performance than the quadratic skipping. 3) Furthermore, we observed that the methods DDIM, SN-DDPM and NPR-DDIM exhibit instability, particularly when a higher percentage of sampling steps is masked. The results deteriorate when 50% of the sampling steps are masked. Additionally, we present the MAE results on Figure 2(a). Our method achieves better MAE results than the baseline with denoising steps by only using 20% denoising steps. Moreover, our method demonstrates greater effectiveness at higher masking rates compared to DDIM (Song et al., 2020a).

## 5.2 ABLATION STUDY

In this section, we aim to provide a comprehensive understanding of EDDPM by showcasing the distribution of probability values $\mathbf{s}_i$'s throughout the complete training process and visually repre-

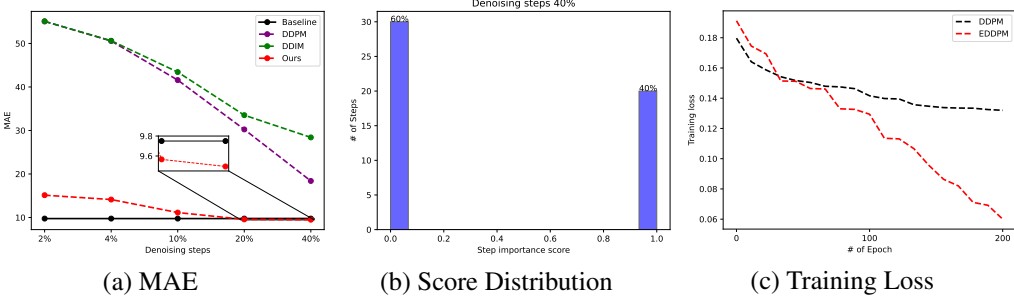

(a) MAE       (b) Score Distribution       (c) Training Loss

Figure 2: a) MAE results, in which our EDDPM method performs better than other DPMs when facing different masking rates. The "Baseline" indicates the imputated samples are generated by 100% sampling steps. b) The histogram of the probability $\mathbf{s}$ learned by our proposed EDDPM, and the results are obtained from Air-quality dataset. Almost all of the $\mathbf{s}_i(i \in \{1, \ldots, T\})$ are either 0 or 1, making $\mathbf{m}$ becomes deterministic. c) The curves of training losses of DDPM and EDDPM. EDDPM converges much more fast than DDPM in the late stage since it identify the informative steps and focus on them to train the weights.

Table 3: Comparing the **MAE** results of DDPM and our proposed EDDPM. "Scratch" presents that the denoising steps are searched from scratch by our EDDPM, "Finetune" uses the pretrained DDPM models to search the most informative steps by finetuning one epoch. Here, DDPM and Ours(Stratch) have the same training time cost since they run for the same number of epoches and the training cost of updating $\mathbf{s}$ is negligible due to the efficient policy gradient estimator.

| Method | Denoising steps | | | | | Training time |
|---|---|---|---|---|---|---|
| | 2% | 4% | 10% | 20% | 40% | |
| DDPM | 55.075 | 50.566 | 41.585 | 30.243 | 18.399 | 4 h |
| Ours (Scratch) | 15.125 | 14.128 | 11.135 | 9.5660 | **9.495** | 4 h |
| Ours (Finetune) | 24.747 | 13.876 | 12.885 | 11.142 | 9.892 | **1 min** |

senting the training progress. Additionally, we demonstrate the effectiveness of EDDPM in model compression, i.e., identifying the optimal denoising trajectory within pre-trained diffusion models.

**Convergence to Deterministic Mask.** Figure 2(b) illustrates the convergence of the probabilities learned by our method after training. It is evident that the probabilities $\mathbf{s}_i$ tend to converge towards either 0 or 1, resulting in a deterministic mask. This characteristic allows us to safely discard the insignificant steps once training is complete. The effectiveness of this convergence can be attributed to the global sparsity constraint imposed on $\mathbf{s}$ as we discussed in Section 4.

**Fast Convergence.** In Figure 2(c), we present a visualization of the impact of EDDPM on improving training efficiency. We can observe that the training loss of EDDPM decreases much faster than DDPM, especially in the late stage. The reason is that as the training goes on most of the $\mathbf{s}_i(i \in \{1, \ldots, T\})$ would get closer to either 0 or 1 (see Figure 2(b)), which enables EDDPM focus on the most informative steps to train the model weights.

**Finetuning Pretrained Diffusion Models.** We assess the effectiveness of EDDPM in compressing diffusion models. Specifically, we employ EDDPM on a pre-trained diffusion model to determine the optimal denoising trajectory. The outcomes presented in Table 3 indicate that EDDPM can achieve comparable performance in only 1 epoch, in contrast to the model trained from scratch. This highlights the potential of EDDPM as a practical tool for exploring large diffusion models with resource constraints.

## 6   CONCLUSION

In this paper, we propose an efficient denoising diffusion model via probabilistic masking to accelerate sample generation process. The main contribution is that the proposed probabilistic masking approach can identity and remove the redundant steps gradually during training. We re-implement several latest sampling acceleration methods on two time series imputation benchmarks and construct experiments on image generation task to verify the effectiveness of the proposed EDDPM. We also find that our method can find the optimal denoising steps by only using one epoch. This makes it possible to explore large diffusion models in academia with limited resources.

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

---

**Algorithm 1** Efficient Denoising Diffusion via Probabilistic Masking (EDDPM)

---

**Require:** Random initilized diffusion model $\mathcal{F}_\theta$, $N$ is the training epoch, masking rate $\gamma_f$.

**Initilization:** Sampling probabilities for each time step: $\mathbf{s} = \mathbf{1} \in \mathbb{R}^T$

 1: **for** epoch $e = 1, 2, .., N$ **do**
 2:     Calculate $\gamma_e$ according to Eqn.(12).
 3:     **for** each training iteration **do**
 4:         Sample mini batch of data $\mathcal{X}_B$.
 5:         Bernoulli sampling based on scores $\mathbf{s}$ for masking diffusion steps.
 6:         Update variance schedule based on the sampled masks with Eqn.(7).
 7:         Random sample the unmasked diffusion step for training.
 8:         Compute diffusion model loss $\mathcal{L}_\theta^t$.
 9:         Back-propagation for $\mathcal{F}_\theta$ to estimate $\nabla_\theta \Phi(\theta, \mathbf{s})$.
10:         Estimate $\nabla_s \Phi(\theta, \mathbf{s})$ according to Eqn. (11) .
11:         Update $\theta$ and $\mathbf{s}$ according to Eqn.(13).
12:     **end for**
13: **end for**

---

## A    DIFFUSION MODEL WITH PROBABILISTIC MASKS

The masked *forward process* is

$$q(\mathbf{x}_t|\mathbf{x}_{t-1}, \mathbf{m}_t) = \mathcal{N}(\mathbf{x}_t; \sqrt{1 - \beta_t \mathbf{m}_t}\mathbf{x}_{t-1}, \beta_t \mathbf{m}_t \mathbf{I});$$
$$q(\mathbf{x}_{1:T}, \mathbf{m}|\mathbf{x}_0) = q(\mathbf{x}_{1:T}|\mathbf{x}_0, \mathbf{m})p_\mathbf{s}(\mathbf{m})$$
$$= p_\mathbf{s}(\mathbf{m})\Pi_{t=1}^T q(\mathbf{x}_t|\mathbf{x}_{t-1}, \mathbf{m}_t)$$
$$= \Pi_{t=1}^T q(\mathbf{x}_t|\mathbf{x}_{t-1}, \mathbf{m}_t)p_\mathbf{s}(\mathbf{m}_t)$$

The masked *reverse process* is

$$p_\theta(\mathbf{x}_{t-1}|\mathbf{x}_t, \mathbf{m}_t) = \mathcal{N}(\mathbf{x}_{t-1}; \mu_\theta(\mathbf{x}_t, \mathbf{m}_t, t), \mathbf{m}_t \Sigma_\theta(\mathbf{x}_t, t))$$
$$p_\theta(\mathbf{x}_{0:T}, \mathbf{m}) = p_\mathbf{s}(\mathbf{m})p_\theta(\mathbf{x}_{0:T}|\mathbf{m})$$
$$= p_\mathbf{s}(\mathbf{m})p(\mathbf{x}_T)\Pi_{t=1}^T p_\theta(\mathbf{x}_{t-1}|\mathbf{x}_t, \mathbf{m}_t)$$
$$= p(\mathbf{x}_T)\Pi_{t=1}^T p_\theta(\mathbf{x}_{t-1}|\mathbf{x}_t, \mathbf{m}_t)p_\mathbf{s}(\mathbf{m}_t)$$

The variational low bound for diffusion models can be written as:

$$-\log p_\theta(\mathbf{x}_0) = -\log \int p_\theta(\mathbf{x}_{0:T}, \mathbf{m}) d\mathbf{x}_{1:T} d\mathbf{m}$$
$$= -\log \int \frac{p_\theta(\mathbf{x}_{0:T}, \mathbf{m})}{q(\mathbf{x}_{1:T}, \mathbf{m}|\mathbf{x}_0)} q(\mathbf{x}_{1:T}, \mathbf{m}|\mathbf{x}_0) d\mathbf{x}_{1:T} d\mathbf{m}$$
$$= -\log \mathbb{E}_q \frac{p_\theta(\mathbf{x}_{0:T}, \mathbf{m})}{q(\mathbf{x}_{1:T}, \mathbf{m}|\mathbf{x}_0)}$$
$$\leq -\mathbb{E}_q \log \frac{p_\theta(\mathbf{x}_{0:T}, \mathbf{m})}{q(\mathbf{x}_{1:T}, \mathbf{m}|\mathbf{x}_0)}$$
$$= -\mathbb{E}_q \log \frac{p(\mathbf{x}_T|\mathbf{m})\Pi_{t=1}^T p_\theta(\mathbf{x}_{t-1}|\mathbf{x}_t, \mathbf{m})p_\mathbf{s}(\mathbf{m}_t)}{\Pi_{t=1}^T q(\mathbf{x}_t|\mathbf{x}_{t-1}, \mathbf{m})p_\mathbf{s}(\mathbf{m}_t)}$$
$$= -\mathbb{E}_q \left[ \log p(\mathbf{x}_T|\mathbf{m}) + \sum_{t=1}^T \log \frac{p_\theta(\mathbf{x}_{t-1}|\mathbf{x}_t, \mathbf{m})}{q(\mathbf{x}_t|\mathbf{x}_{t-1}, \mathbf{m})} \right] =: \mathcal{L}$$

$$
\begin{aligned}
q(\mathbf{x}_t|\mathbf{x}_{t-1}, \mathbf{m}) &= q(\mathbf{x}_t|\mathbf{x}_0, \mathbf{x}_{t-1}, \mathbf{m}) \\
&= \frac{q(\mathbf{x}_t, \mathbf{x}_0, \mathbf{x}_{t-1}, \mathbf{m})}{q(\mathbf{x}_0, \mathbf{x}_{t-1}, \mathbf{m})} \\
&= \frac{q(\mathbf{x}_{t-1}|\mathbf{x}_t, \mathbf{x}_0, \mathbf{m})q(\mathbf{x}_t, \mathbf{x}_0, \mathbf{m})}{q(\mathbf{x}_0, \mathbf{x}_{t-1}, \mathbf{m})} \\
&= \frac{q(\mathbf{x}_{t-1}|\mathbf{x}_t, \mathbf{x}_0, \mathbf{m})q(\mathbf{x}_t|\mathbf{x}_0, \mathbf{m})q(\mathbf{x}_0, \mathbf{m})}{q(\mathbf{x}_{t-1}|\mathbf{x}_0, \mathbf{m})q(\mathbf{x}_0, \mathbf{m})} \\
&= q(\mathbf{x}_{t-1}|\mathbf{x}_t, \mathbf{x}_0, \mathbf{m})\frac{q(\mathbf{x}_t|\mathbf{x}_0, \mathbf{m})}{q(\mathbf{x}_{t-1}|\mathbf{x}_0, \mathbf{m})}
\end{aligned}
$$

$$
\begin{aligned}
\mathcal{L} &= -\mathbb{E}_q\left[\log p(\mathbf{x}_T|\mathbf{m}) + \sum_{t=1}^{T}\log\frac{p_\theta(\mathbf{x}_{t-1}|\mathbf{x}_t, \mathbf{m})}{q(\mathbf{x}_t|\mathbf{x}_{t-1}, \mathbf{m})}\right] \\
&= -\mathbb{E}_q\left[\log p(\mathbf{x}_T|\mathbf{m}) + \sum_{t=2}^{T}\log\frac{p_\theta(\mathbf{x}_{t-1}|\mathbf{x}_t, \mathbf{m})}{q(\mathbf{x}_t|\mathbf{x}_{t-1}, \mathbf{m})} + \log\frac{p_\theta(\mathbf{x}_0|\mathbf{x}_1, \mathbf{m})}{q(\mathbf{x}_1|\mathbf{x}_0, \mathbf{m})}\right] \\
&= -\mathbb{E}_q\left[\log p(\mathbf{x}_T|\mathbf{m}) + \sum_{t=2}^{T}\log\frac{p_\theta(\mathbf{x}_{t-1}|\mathbf{x}_t, \mathbf{m})}{q(\mathbf{x}_{t-1}|\mathbf{x}_t, \mathbf{x}_0, \mathbf{m})}\cdot\frac{q(\mathbf{x}_{t-1}|\mathbf{x}_0, \mathbf{m})}{q(\mathbf{x}_t|\mathbf{x}_0, \mathbf{m})} + \log\frac{p_\theta(\mathbf{x}_0|\mathbf{x}_1, \mathbf{m})}{q(\mathbf{x}_1|\mathbf{x}_0, \mathbf{m})}\right] \\
&= -\mathbb{E}_q\left[\log\frac{p(\mathbf{x}_T|\mathbf{m})}{q(\mathbf{x}_T|\mathbf{x}_0, \mathbf{m})} + \sum_{t=2}^{T}\log\frac{p_\theta(\mathbf{x}_{t-1}|\mathbf{x}_t, \mathbf{m})}{q(\mathbf{x}_{t-1}|\mathbf{x}_t, \mathbf{x}_0, \mathbf{m})} + \log p_\theta(\mathbf{x}_0|\mathbf{x}_1, \mathbf{m})\right] \\
&= \mathbb{E}_q\Bigg[\underbrace{D_{\mathrm{KL}}\left(q(\mathbf{x}_T|\mathbf{x}_0, \mathbf{m})\|p(\mathbf{x}_T|\mathbf{m})\right)}_{\mathcal{L}^T} + \sum_{t=2}^{T}\underbrace{D_{\mathrm{KL}}\left(q(\mathbf{x}_{t-1}|\mathbf{x}_t, \mathbf{x}_0, \mathbf{m})\|p_\theta(\mathbf{x}_{t-1}|\mathbf{x}_t, \mathbf{m})\right)}_{\mathcal{L}^{t-1}} - \log p_\theta(\mathbf{x}_0|\mathbf{x}_1, \mathbf{m})\Bigg]
\end{aligned}
$$

Note that

$$
q(\mathbf{x}_{t-1}|\mathbf{x}_t, \mathbf{x}_0, \mathbf{m}) = \mathcal{N}(\mathbf{x}_{t-1}; \tilde{\mu}(\mathbf{x}_t, \mathbf{x}_0), \tilde{\beta}_t\mathbf{I})
$$

$$
\tilde{\mu}_t(\mathbf{x}_t, \mathbf{x}_0) = \frac{\sqrt{\bar{\alpha}_{t-1}(\mathbf{m})}\beta_t\mathbf{m}_t}{1-\alpha_t(\mathbf{m})}\mathbf{x}_0 + \frac{\sqrt{\alpha_t(\mathbf{m})}(1-\bar{\alpha}_{t-1}(\mathbf{m}))}{1-\bar{\alpha}_t(\mathbf{m})}\mathbf{x}_t,
$$

$$
\tilde{\beta}_t = \frac{1-\bar{\alpha}_{t-1}(\mathbf{m})}{1-\bar{\alpha}_t(\mathbf{m})}\mathbf{m}_t\beta_t.
$$

where

$$
\alpha_t(\mathbf{m}) = 1 - \mathbf{m}_t\beta_t \text{ and } \bar{\alpha}_t(\mathbf{m}) = \Pi_{i=1}^{t}\alpha_i(\mathbf{m}).
$$

For the reverse process, we have

$$
p_\theta(\mathbf{x}_{t-1}|\mathbf{x}_t, \mathbf{m}) = \mathcal{N}(\mathbf{x}_{t-1}; \mu_\theta(\mathbf{x}_t, \mathbf{m}, t), \tilde{\sigma}_t^2(\mathbf{m})\mathbf{I}).
$$

We define

$$
\delta(\tilde{\mu}_t, \mu_\theta) = \frac{1}{\tilde{\sigma}_t^2(\mathbf{m})}\|\tilde{\mu}_t(\mathbf{x}_t, \mathbf{x}_0) - \mu_\theta(\mathbf{x}_t, \mathbf{m}, t)\|^2,
$$

we can get,

$$
\begin{aligned}
\mathcal{L}^{t-1} &= \begin{cases} 0, & \text{if } \mathbf{m}_t = 0 \\ \frac{1}{2}\left[n\frac{1-\bar{\alpha}_{t-1}(\mathbf{m})}{1-\bar{\alpha}_t(\mathbf{m})}\frac{\mathbf{m}_t\beta_t}{\tilde{\sigma}_t^2(\mathbf{m})} - n + \delta(\tilde{\mu}_t, \mu_\theta) + n\log\left(\frac{1-\bar{\alpha}_{t-1}(\mathbf{m})}{1-\bar{\alpha}_t(\mathbf{m})}\frac{\mathbf{m}_t\beta_t}{\tilde{\sigma}_t^2(\mathbf{m})}\right)\right], & \text{otherwise} \end{cases} \\
&= \begin{cases} 0, & \text{if } \mathbf{m}_t = 0 \\ \frac{1}{2}\delta(\tilde{\mu}_t, \mu_\theta) + \frac{n}{2}\left[\frac{1-\bar{\alpha}_{t-1}(\mathbf{m})}{1-\bar{\alpha}_t(\mathbf{m})}\frac{\mathbf{m}_t\beta_t}{\tilde{\sigma}_t^2(\mathbf{m})} - 1 + \log\left(\frac{1-\bar{\alpha}_{t-1}(\mathbf{m})}{1-\bar{\alpha}_t(\mathbf{m})}\frac{\mathbf{m}_t\beta_t}{\tilde{\sigma}_t^2(\mathbf{m})}\right)\right], & \text{otherwise} \end{cases} \\
&= \begin{cases} 0, & \text{if } \mathbf{m}_t = 0 \\ \frac{1}{2}\delta(\tilde{\mu}_t, \mu_\theta) + C(\mathbf{m}) & \text{otherwise} \end{cases}
\end{aligned}
$$

where

$$C(\mathbf{m}) = \frac{n}{2}\left[\frac{1-\bar{\alpha}_{t-1}(\mathbf{m})}{1-\bar{\alpha}_t(\mathbf{m})}\frac{\mathbf{m}_t\beta_t}{\tilde{\sigma}_t^2(\mathbf{m})} - 1 + \log\left(\frac{1-\bar{\alpha}_{t-1}(\mathbf{m})}{1-\bar{\alpha}_t(\mathbf{m})}\frac{\mathbf{m}_t\beta_t}{\tilde{\sigma}_t^2(\mathbf{m})}\right)\right].$$

In this paper, following DDPM, we choose

$$\tilde{\sigma}_t^2(\mathbf{m}) = \frac{1-\bar{\alpha}_{t-1}(\mathbf{m})}{1-\bar{\alpha}_t(\mathbf{m})}\mathbf{m}_t\beta_t.$$

In this case,

$$C(\mathbf{m}) = 0.$$

For $\tilde{\mu}_t(\mathbf{x}_t, \mathbf{x}_0)$, since

$$\mathbf{x}_t(\mathbf{x}_0, \epsilon) = \sqrt{\bar{\alpha}_t(\mathbf{m})}\mathbf{x}_0 + \sqrt{1-\bar{\alpha}_t(\mathbf{m})}\epsilon \text{ with } \epsilon \sim \mathcal{N}(\mathbf{0}, \mathbf{I}),$$

we have

$$\begin{aligned}
\tilde{\mu}_t(\mathbf{x}_t, \mathbf{x}_0) &= \frac{\sqrt{\bar{\alpha}_{t-1}(\mathbf{m})}\mathbf{m}_t\beta_t}{1-\bar{\alpha}_t(\mathbf{m})}\mathbf{x}_0 + \frac{\sqrt{\alpha_t(\mathbf{m})}(1-\bar{\alpha}_{t-1}(\mathbf{m}))}{1-\bar{\alpha}_t(\mathbf{m})}\mathbf{x}_t(\mathbf{x}_0, \epsilon) \\
&= \frac{\sqrt{\bar{\alpha}_{t-1}(\mathbf{m})}\mathbf{m}_t\beta_t}{1-\bar{\alpha}_t(\mathbf{m})}\frac{1}{\sqrt{\bar{\alpha}_t(\mathbf{m})}}\left(\mathbf{x}_t(\mathbf{x}_0, \epsilon) - \sqrt{1-\bar{\alpha}_t(\mathbf{m})}\epsilon\right) \\
&\quad + \frac{\sqrt{\alpha_t(\mathbf{m})}(1-\bar{\alpha}_{t-1}(\mathbf{m}))}{1-\bar{\alpha}_t(\mathbf{m})}\mathbf{x}_t(\mathbf{x}_0, \epsilon) \\
&= \frac{1}{\sqrt{\alpha_t(\mathbf{m})}}\left(\mathbf{x}_t(\mathbf{x}_0, \epsilon) - \frac{\mathbf{m}_t\beta_t}{\sqrt{1-\bar{\alpha}_t(\mathbf{m})}}\epsilon\right).
\end{aligned}$$

Hence, we define

$$\mu(\mathbf{x}_t, \mathbf{m}, t) = \frac{1}{\sqrt{\alpha_t(\mathbf{m})}}\left(\mathbf{x}_t - \frac{\mathbf{m}_t\beta_t}{\sqrt{1-\bar{\alpha}_t(\mathbf{m})}}\epsilon_\theta(\mathbf{x}_t, t)\right).$$

Then, we have

$$\begin{aligned}
&\frac{1}{2\tilde{\sigma}_t^2(\mathbf{m})}\left\|\tilde{\mu}_t(\mathbf{x}_t, \mathbf{x}_0) - \mu_\theta(\mathbf{x}_t, \mathbf{m}, t)\right\|^2 \\
&= \frac{\mathbf{m}_t\beta_t^2}{2\tilde{\sigma}_t^2(\mathbf{m})\alpha_t(\mathbf{m})(1-\bar{\alpha}_t(\mathbf{m}))}\left\|\epsilon - \epsilon_\theta\left(\mathbf{x}_t, t\right)\right\|^2 \\
&= \frac{\mathbf{m}_t\beta_t^2}{2\tilde{\sigma}_t^2(\mathbf{m})\alpha_t(\mathbf{m})(1-\bar{\alpha}_t(\mathbf{m}))}\left\|\epsilon - \epsilon_\theta\left(\sqrt{\bar{\alpha}_t(\mathbf{m})}\mathbf{x}_0 + \sqrt{1-\bar{\alpha}_t(\mathbf{m})}\epsilon, t\right)\right\|^2
\end{aligned}$$

Finally, we get the loss as follows:

$$\mathcal{L}^{t-1} = \begin{cases} 0, & \text{if } \mathbf{m}_t = 0 \\ \frac{\mathbf{m}_t\beta_t^2}{2\tilde{\sigma}_t^2(\mathbf{m})\alpha_t(\mathbf{m})(1-\bar{\alpha}_t(\mathbf{m}))}\left\|\epsilon - \epsilon_\theta\left(\sqrt{\bar{\alpha}_t(\mathbf{m})}\mathbf{x}_0 + \sqrt{1-\bar{\alpha}_t(\mathbf{m})}\epsilon, t\right)\right\|^2, & \text{otherwise} \end{cases}$$

Thus, we get the objective function in the main paper.

## B  PROJECTION OPERATOR IMPLEMENTATION

**Theorem 1.** *Given a vector $\mathbf{z}$, its projection $\mathbf{s}$ onto our constraint region $\{\mathbf{s} \in \mathbb{R}^T : \|\mathbf{s}\|_1 \leq K_e, \mathbf{s} \in [0, 1]^T\}$ can be computed as follows:*

$$\mathbf{s} = \min(1, \max(0, \mathbf{z} - v_2^*\mathbf{1})).$$

*where $v_2^* = \max(0, v_1^*)$ and $v_1^*$ is the solution to the following equation:*

$$\mathbf{1}^\top[\min(1, \max(0, \mathbf{z} - v_1^*\mathbf{1}))] - K_e = 0. \tag{14}$$

The equation (14) can be efficiently solved using the bisection method.

We would like to point out that the theorem above as well as its proof is standard and similar cases can be found in (Wang & Carreira-Perpinán, 2013). To make this paper self-contained, we present them in the appendix, although this is not our contribution.

*Proof.* The projection of $\mathbf{z}$ onto the set $\{\mathbf{s} \in \mathbb{R}^T : \|\mathbf{s}\|_1 \leq K_e, \mathbf{s} \in [0, 1]^T\}$ can be formulated as the following optimization problem:

$$\min_{\mathbf{s} \in \mathbb{R}^n} \frac{1}{2}\|\mathbf{s} - \mathbf{z}\|^2,$$
$$s.t. \mathbf{1}^\top \mathbf{s} \leq K_e \text{ and } 0 \leq s_i \leq 1, i = 1, \ldots, T.$$

We derive the solution as follows.

The Lagrangian of the problem is given by:

$$L(\mathbf{s}, v) = \frac{1}{2}\|\mathbf{s} - \mathbf{z}\|^2 + v(\mathbf{1}^\top \mathbf{s} - K_e) \tag{15}$$
$$= \frac{1}{2}\|\mathbf{s} - (\mathbf{z} - v\mathbf{1})\|^2 + v(\mathbf{1}^\top \mathbf{z} - K_e) - \frac{n}{2}v^2. \tag{16}$$

subject to $v \geq 0$ and $0 \leq \mathbf{s}_i \leq 1$.

Minimizing the problem with respect to $\mathbf{s}$, we obtain:

$$\tilde{\mathbf{s}} = \mathbf{1}_{\mathbf{z} - v\mathbf{1} \geq 1} + (\mathbf{z} - v\mathbf{1})_{1 > \mathbf{z} - v\mathbf{1} > 0} \tag{17}$$

Then we have:

$$g(v) = L(\tilde{\mathbf{s}}, v)$$
$$= \frac{1}{2}\|[\mathbf{z} - v\mathbf{1}]_- + [\mathbf{z} - (v + 1)\mathbf{1}]_+\|^2$$
$$+ v(\mathbf{1}^\top \mathbf{z} - K_e) - \frac{n}{2}v^2$$
$$= \frac{1}{2}\|[\mathbf{z} - v\mathbf{1}]_-\|^2 + \frac{1}{2}\|[\mathbf{z} - (v + 1)\mathbf{1}]_+\|^2$$
$$+ v(\mathbf{1}^\top \mathbf{z} - K_e) - \frac{n}{2}v^2, v \geq 0.$$
$$g'(v) = \mathbf{1}^\top[v\mathbf{1} - \mathbf{z}]_+ + \mathbf{1}^\top[(v + 1)\mathbf{1} - \mathbf{z}]_-$$
$$+ (1^T\mathbf{z} - K_e) - nv$$
$$= \mathbf{1}^\top \min(1, \max(0, \mathbf{z} - v\mathbf{1})) - K_e, v \geq 0.$$

To verify that $g'(v)$ is a monotone decreasing function with respect to $v$, we can use the bisection method to solve the equation $g'(v) = 0$ and find the solution $v_1^*$. It can be observed that $g(v)$ increases in the range $(-\infty, v_1^*]$ and decreases in the range $[v_1^*, +\infty)$. The maximum of $g(v)$ is achieved at 0 if $v_1^* \leq 0$, and at $v_1^*$ if $v_1^* > 0$. We then set $v_2^* = max(0, v_1^*)$. Finally, the projection $\mathbf{s}^*$ is given by:

$$\mathbf{s}^* = \mathbf{1}_{\mathbf{z} - v_2^*\mathbf{1} \geq 1} + (\mathbf{z} - v_2^*\mathbf{1})_{1 > \mathbf{z} - v_2^*\mathbf{1} > 0} \tag{18}$$
$$= \min(1, \max(0, \mathbf{z} - v_2^*\mathbf{1})). \tag{19}$$

$\square$

## C  ANALYSIS ON THE SPARSITY OF THE OPTIMAL SCORE

As we claimed in the main text, most the elements of score vector $\mathbf{s}$ would converge to either 0 or 1 as the training goes on. It is equivalent to say that most of the elements in the optimal solution $\mathbf{s}^*$ are either 0 or 1. To explain this, let us start from understanding the sparsity of lasso.

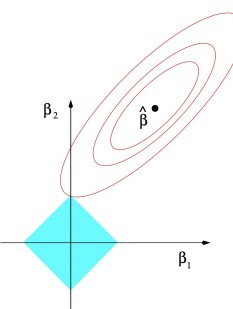

Figure 3: Estimation picture for the lasso (Hastie et al., 2015).

Prof. Robert Tibshirani, the author of the well-known sparse learning method lasso, provides an explanation on the sparsity in lasso from a geometric perspective in pages 10-12 of his book titled Statistical Learning with Sparsity: The Lasso and Generalizations (Hastie et al., 2015). To be precise, the optimization problem of lasso is equivalent to the following one with some $t$:

$$\min_{\boldsymbol{\beta}} \|\mathbf{y} - \mathbf{X}\boldsymbol{\beta}\|^2, \text{ s.t. } \sum_{i=1}^{p} |\beta_i| \leq t,$$

where $\mathbf{X} \in \mathbb{R}^{n \times p}$ is the feature matrix of $n$ samples and $\mathbf{y} \in \mathbb{R}^n$ is the response vector. Note that the constraint region above is a diamond ($p = 2$) or a rhomboid ($p > 2$). As shown in Figure 3, which is copied from page 11 of the above textbook, the optimal solution is the point, where the elliptical contours of the loss hit this constraint region. When the dimension $p = 2$, the diamond has corners; if the solution occurs at a corner, then it has one parameter $\beta_j$ equal to 0. When $p > 2$, the diamond becomes a rhomboid, and has many corners, flat edges, and faces; there are many more opportunities for the estimated parameters to be zero. Please refer to page 12 of the above book for more details.

The situation in our problem is essentially the same with lasso, the only difference is that our constraint region $\{\sum_{t=1}^{T} |s_t| \leq K, \mathbf{s} \in [0,1]^T\}$ has more corners (i.e., the coordinates are 0 or 1) than that of lasso, therefore, the optimal $s_t$ has a high probability to be either 0 or 1.

# D BASICS

## D.1 MUTUAL SKIPPING OF SAMPLING STEPS

To improve the efficiency of sample generation process, previous methods (Song et al., 2020a; Bao et al., 2022; 2021) always manually select the denoising steps through uniform skipping and quadratic skipping. The mathematical expression of the above skipping approaches can be written as:

$$\mathbb{T} = \{1, 1 + S, ..., 1 + iS, ..., L\}, \text{ with } S = \begin{cases} \frac{T}{L} & , \text{ uniform skipping,} \\ \left(\frac{0.8T}{L}\right)^2 & , \text{ quadratic skipping.} \end{cases} \quad (20)$$

where $i = 1, \ldots, L$. $T$ and $L$ are the number of diffusion steps and number of denoising steps in the training and testing state respectively. $S$ is the skipping step. The difference of $T$ and $L$ results in decoupled forward and reverse processes, which makes a suboptimal performance. Instead, our proposed probabilistic masking method can identify and keep the most informative steps during training.

## D.2 MULTIVARIATE TIME SERIES IMPUTATION

Let us denote each time series as $\mathbf{X} \in R^{K \times P}$, where $K$ is the number of features and $P$ is the length of time series. Probabilistic time series imputation is to estimate the missing values of $X$ by

exploiting the observed values of $X$. The diffusion model is used to estimate the true conditional data distribution $q(x_0^t|x_0^c)$, where $x_0^t$ and $x_0^c$ are the imputation targets and conditional observations respectively.

# E    EXPERIMENTS

## E.1    TIME SERIES DATASETS

Healthcare dataset (Silva et al., 2012) consists of 4000 clinical time series with 35 variables for 48 hours from intensive care unit (ICU), and it contains around 80% missing values. Following previous study (Tashiro et al., 2021), we randomly choose 10/50/90% of observed values as ground-truth on the test data for imputation.

Air-quality dataset is composed of air quality data from 36 stations in Beijing from 2014/05/01 to 2015/04/30, and it has around 13% missing values. We set 36 consecutive time steps as one time series. To build missing values in the time series, we follow the empirical settings of the baseline (Tashiro et al., 2021), we adopt the random strategy for the healthcare dataset and the mix of the random and historical strategy for the air quality dataset.

## E.2    IMPLEMENT DETAILS

All the experiments are implemented by Pytorch 1.7.0 on a virtual workstation with 8 11G memory Nvidia GeForce RTX 2080Ti GPUs.

**Time series.** As for model hyper-parameters, we set the batch size as 16 and the number of epochs as 200. We used Adam (Kingma & Ba, 2014) optimizer with learning rate 0.001 that is decayed to 0.0001 and 0.00001 at 75% and 90% of the total epochs, respectively. For the diffusion model, we follow the CSDI (Tashiro et al., 2021) architecture to set the number of residual layers as 4, residual channels as 64, and attention heads as 8. The denoising step $T$ is set to 50 as our baseline.

**Image data.** Following (Nichol & Dhariwal, 2021), we use the U-Net model architecture, train 500K iterations with a batch size of 128, use a learning rate of 0.0001 with the Adam (Kingma & Ba, 2014) optimizer and use an exponential moving average (EMA) with a rate of 0.9999. The denoising step $T$ is set to 1000 and the linear forward noise schedule is used as our baseline.

## E.3    EVALUATION METRIC

The detailed formulations of three metrics for time series task are:

$$MAE(x, \hat{x}) = \frac{1}{N} \sum_{i=1}^{N} \|x_i - \hat{x}_i\|, \tag{21}$$

$$RMSE(x, \hat{x}) = \sqrt{\frac{1}{N} \sum_{i=1}^{N} \|x_i - \hat{x}_i\|^2}, \tag{22}$$

$$CRPS(F, \hat{F}) = \int_{-\infty}^{\infty} \left[F(z) - \hat{F}(z)\right]^2 dz, \tag{23}$$

where $x$ denotes the ground truth of the missed time series, $\hat{x}$ represents the predicted values. $F$ is the cumulative distribution function of observations.

## E.4    MAIN RESULTS

As shown in Table 4, our proposed EDDPM can achieve better results than the baselines with 100% steps (blue text) even if $60\% \sim 75\%$ denoising steps are masked. These results are consistent with the conclusion of the main paper.

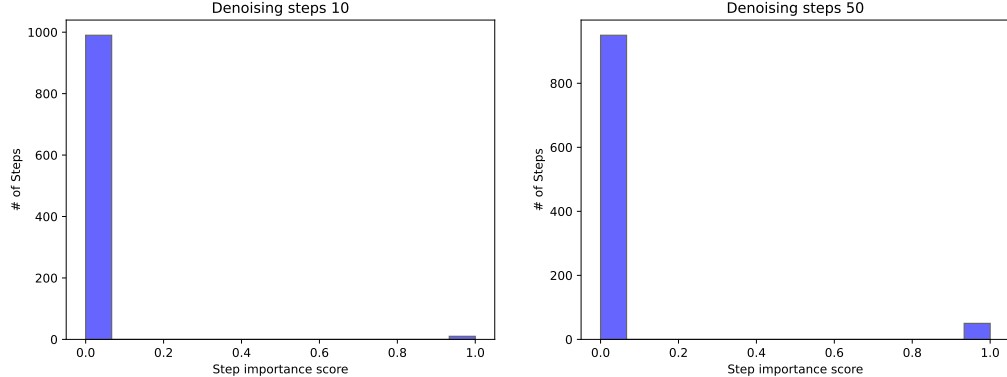

Figure 4: The histogram of the probability **s** learned by our proposed EDDPM, and the results are obtained from CIFAR-10 dataset.

### E.5 VISUALIZATION RESULTS

From the results illustrated in Figure 5, we can conclude that our proposed EDDPM can generate more accurate probabilistic imputation results by only using the original $20\% \sim 50\%$ steps.

For CIFAR-10 image generation, Figure 6 and 8 show that our proposed EDDPM can generate more high-quality image samples than DDIM (Song et al., 2020a) when using 10 denoising steps. Figure 7 and 9 show the sample pairs generated by our EDDPM with 5, 10 and 100 denoising steps, from these results we can conclude that our method generate high-quality CIFAR-10 images using 5 steps.

Table 4: Comparising sampling acceleration methods in terms of **CRPS** results on variable denoising steps. [†] indicate that the sampling is accelerated by quadratic skipping during inference, the others utilize uniform skipping. We highlight the best results that surpass the baselines in red color, which means our method generates high-quality time series with fewer denoising steps. The **bold** results show that our proposed EDDPM achieves better performance than other sampling acceleration methods.

| Dataset | Missing | Method | Denoising steps | | | | Baselines |
|---|---|---|---|---|---|---|---|
| | | | 10% | 25% | 40% | 50% | |
| Healthcare | 10% | DDPM[†] | 0.688 | 0.501 | 0.382 | 0.326 | |
| | | DDPM | 0.640 | 0.431 | 0.344 | 0.276 | |
| | | DDIM | 0.641 | 0.495 | 0.564 | 0.840 | |
| | | AnalyticDPM | 0.615 | 0.536 | 0.516 | 0.501 | 0.238 |
| | | SN-DDPM | 0.769 | 0.757 | 0.762 | 0.769 | |
| | | NPR-DDIM | 0.573 | 0.502 | 0.504 | 0.516 | |
| | | Ours | **0.267** | 0.237 | 0.235 | 0.231 | |
| | 50% | DDPM[†] | 0.699 | 0.582 | 0.490 | 0.439 | |
| | | DDPM | 0.675 | 0.516 | 0.437 | 0.372 | |
| | | DDIM | 0.675 | 0.562 | 0.601 | 0.810 | |
| | | AnalyticDPM | 0.698 | 0.586 | 0.572 | 0.579 | 0.331 |
| | | SN-DDPM | 0.761 | 0.752 | 0.759 | 0.772 | |
| | | NPR-DDIM | 0.612 | 0.546 | 0.547 | 0.561 | |
| | | Ours | **0.357** | **0.337** | 0.321 | 0.330 | |
| | 90% | DDPM [†] | 0.731 | 0.690 | 0.648 | 0.622 | |
| | | DDPM | 0.737 | 0.654 | 0.594 | 0.557 | |
| | | DDIM | 0.737 | 0.695 | 0.715 | 0.856 | |
| | | AnalyticDPM | 0.715 | 0.685 | 0.672 | 0.668 | 0.522 |
| | | SN-DDPM | 0.840 | 0.810 | 0.808 | 0.810 | |
| | | NPR-DDIM | 0.704 | 0.647 | 0.643 | 0.644 | |
| | | Ours | **0.572** | 0.517 | 0.516 | 0.513 | |
| | | | 4% | 10% | 20% | 40% | |
| Air-quality | 13% | DDPM[†] | 0.568 | 0.482 | 0.374 | 0.217 | |
| | | DDPM | 0.536 | 0.453 | 0.344 | 0.209 | |
| | | DDIM[†] | 0.569 | 0.507 | 0.464 | 0.605 | |
| | | DDIM | 0.537 | 0.485 | 0.619 | 1.553 | |
| | | AnalyticDPM | 0.489 | 0.453 | 0.429 | 0.382 | 0.109 |
| | | SN-DDPM | 0.557 | 0.507 | 0.482 | 0.481 | |
| | | SN-DDIM | 0.653 | 0.568 | 0.558 | 0.582 | |
| | | NPR-DDPM | 0.359 | 0.355 | 0.377 | 0.395 | |
| | | NPR-DDIM | 0.362 | 0.344 | 0.305 | 0.271 | |
| | | Ours | **0.170** | **0.133** | **0.112** | 0.104 | |

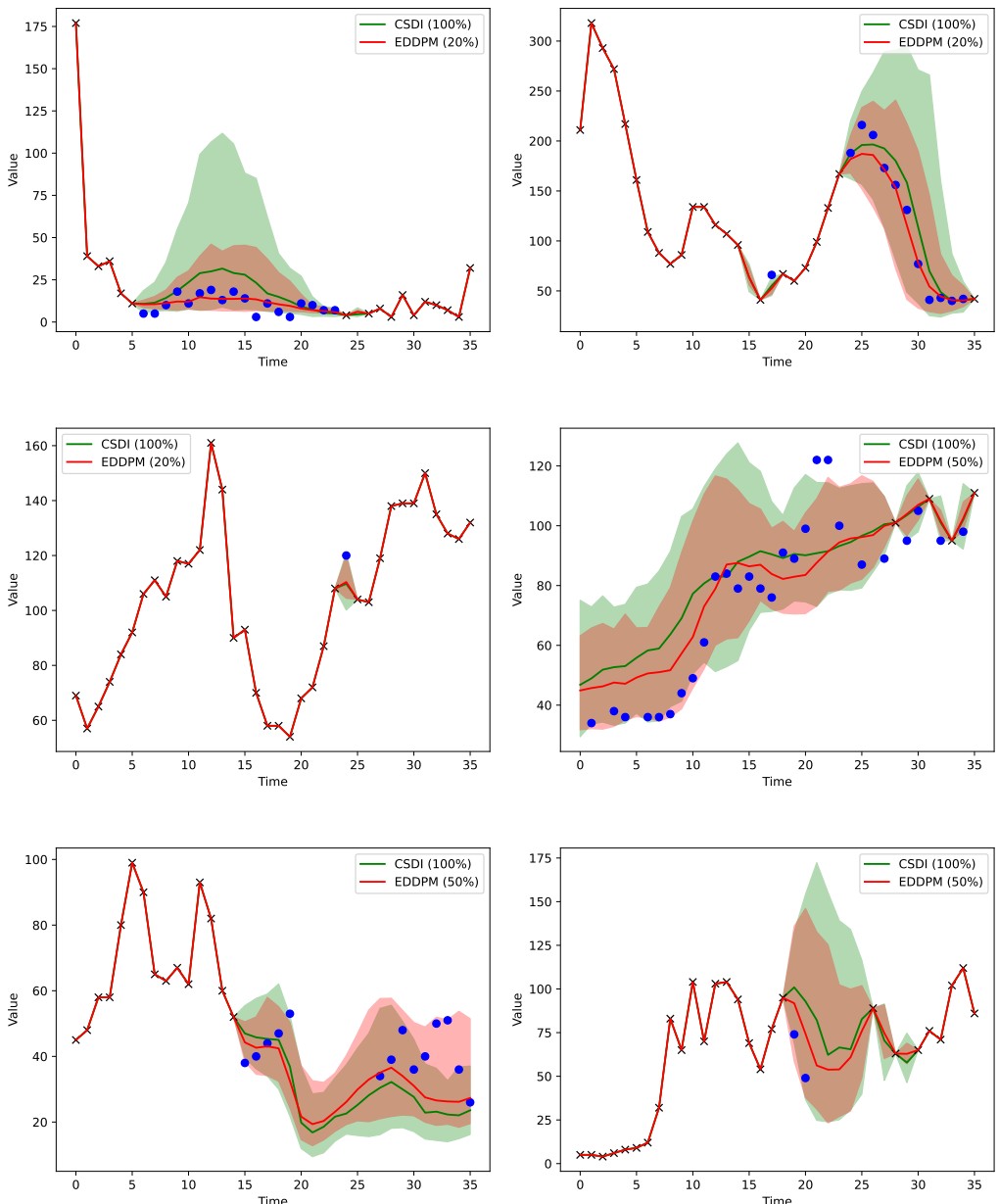

Figure 5: The comparison of our EDDPM method and DDPM Ho et al. (2020) for probabilistic time series imputation on Air-quality dataset. CSDI model is trained by DDPM. The black crosses show observed values and the blue circles show ground-truth imputation targets. red and green colors correspond to our EDDPM and CSDI, respectively. For each method, median values of imputations are shown as the line and 5% and 95% quantiles are shown as the shade.

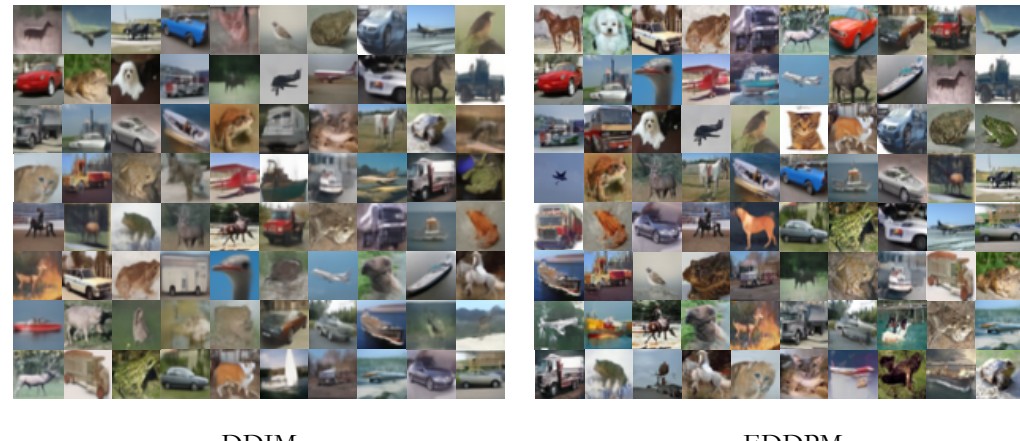

DDIM                                                    EDDPM

Figure 6: Random samples generated by DDIM Song et al. (2020a) and EDDPM (ours) with 10 denoising steps on CIFAR-10 dataset. We only present the result in this extreme sparse case since the results for more denoising steps are difficult to differentiate for human beings.

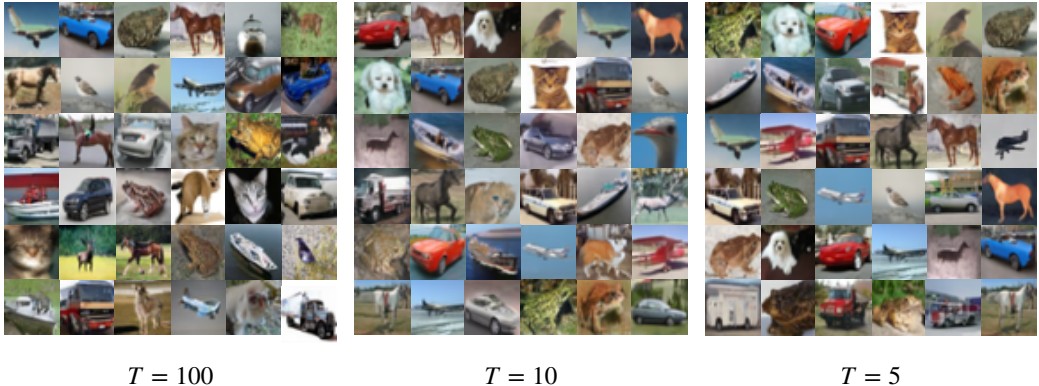

$T = 100$                        $T = 10$                        $T = 5$

Figure 7: Random samples generated by our EDDPM with 5, 10 and 100 denoising steps on CIFAR-10 dataset.

DDIM

EDDPM

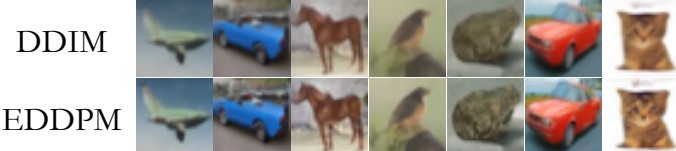

Figure 8: Sample pair comparison based on DDIM Song et al. (2020a) and EDDPM (ours) with 10 denoising steps on CIFAR-10 dataset. We can see that our method can generate images with more details.

$T = 5$

$T = 10$

$T = 100$

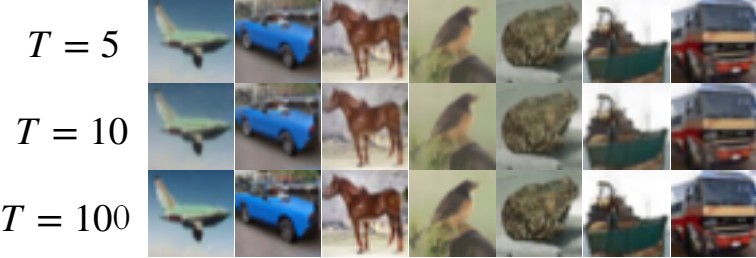

Figure 9: Random samples generated by our EDDPM with 5, 10 and 100 denoising steps on CIFAR-10 dataset.

