# OpenReview forum: "Efficient Denoising Diffusion via Probabilistic Masking"
_ICLR.cc/2024/Conference — ICLR 2024 Conference Withdrawn Submission_

### Official Review · Reviewer_jbL3 · 2023-10-26

**Soundness:** 3 good
**Presentation:** 2 fair
**Contribution:** 3 good
**Rating:** 6
**Confidence:** 4

**Summary:**

This work introduced an Efficient Denoising Diffusion method via Probabilistic Masking (EDDPM) to address the computational intensity issue in diffusion models. EDDPM utilizes probabilistic reparameterization to determine whether a time step should be skipped or not, thereby identifying and eliminating redundant steps during training. This approach, which jointly learns mask distribution parameters with model weights, includes a real-time sparse constraint to significantly enhance training efficiency. Remarkably, as the model reaches full proficiency, random masks converge to a sparse and deterministic form, retaining only crucial steps.

**Strengths:**

The paper introduces the concept of an Efficient Denoising Diffusion Model (EDDPM) to improve the sampling efficiency. This innovation addresses a significant challenge in diffusion models, which often require a large number of steps for generating a single sample. EDDPM effectively identifies and skips redundant steps, enhancing the sampling process. In my eyes, this idea is interesting.

**Weaknesses:**

This article seems to be insufficiently prepared, containing various typos and using somewhat inappropriate notation, which can be confusing for readers. Additionally, the paper lacks intuitive explanations for some of the conclusions, and I will detail these issues in the following question.

**Questions:**

1. As far as I know, there is typically a trade-off between sampling speed and sample quality. Having fewer sampling steps usually improves the sampling speed but often results in a decline in the quality of generated samples. This observation has been discussed in numerous studies focused on accelerating diffusion sampling. Why does Figure 1 indicate an enhancement in sample quality when the number of sampling steps is very low? Is this a consequence of randomness or an mean outcome? Is there a qualitative explanation?

2. If I didn't miss it, the article doesn't employ the L0 norm. If that's the case, I recommend not introducing L0 in the notation section. Additionally, in Equation 6 and subsequent formulas, if you are using the L2 norm, I suggest explicitly writing it as $||\cdot||_2$ not $||\cdot||$.
3. Under section 3, "In this section, for the convenience of presenting our method DDPM......", I think it should be EDDPM;
4. Regarding the mask variable, $\mathbf{m}_t$, I understand it to be the $t$-th entry of the vector $\mathbf{m}$. Following tradition, a single random variable should not be represented in bold form, and it is recommended to write as $m_t$.
5. Under Eq.(9), $\tilde{\boldsymbol{\mu}}(\mathbf{x}_t,\mathbf{x}_0)$, $\tilde{\beta}_t$ should be written as $\tilde{\boldsymbol{\mu}}_t(\mathbf{x}_t,\mathbf{x}_0,\mathbf{m})$, $\tilde{\beta}_t(\mathbf{m})$, because they depend on $\mathbf{m}$.
6. Under Eq.(9), in the definition of $\tilde{\beta}_t$ it has an extra "t" in the lower right corner.
7. Due to the L1 constraint on $\mathbf{s}$, most entries in $\mathbf{s}$ will be pushed to 0. But why are the other entries pushed towards 1? Are there situations where some $m_t$ are around 0.5? In such cases, how should step $t$ be handled? Is there any explanation?
8. Under eq.11, it should be $\nabla_{\theta}\Phi(\theta,s)$ and $\nabla_{s}\Phi(\theta,s)$.
9. Equation 12 is confusing because $\gamma_e$ is not used in the algorithm. Is this a typo? Is it the case that $K=\gamma_e*T$, and $\gamma_e$ is expressed as in equation 12?
10. In the context of image synthesis, the baseline comparison is limited. There are many acceleration sampling algorithms proposed for image synthesis, such as DPM-solver (Lu et al., 2022). While these works are mentioned in the related work section, they are not compared in the experiments.
11. There are several issues with the citation format and some references are duplicated or inappropriately cited in arXiv format, even though they have been published.

---

> ### Author Response · Authors · 2023-11-16
> **Response to Reviewer jbL3 [1]**
>
> Thanks for your constructive comments.
>
> **Q1: As far as I know, there is typically a trade-off between sampling speed and sample quality. Having fewer sampling steps usually improves the sampling speed but often results in a decline in the quality of generated samples. This observation has been discussed in numerous studies focused on accelerating diffusion sampling. Why does Figure 1 indicate an enhancement in sample quality when the number of sampling steps is very low? Is this a consequence of randomness or an mean outcome? Is there a qualitative explanation?**
>
> **A1:** It is a misunderstanding. The x-axis in Figure 1 means the index of the removed step (e.g., 1-st step, 2-nd step), instead of the number of removed steps. We give this result to highlight redundant sampling steps in the denoising process. Removing these steps has a negligible effect on the quality of the samples or may even improve the sample quality.
>
>
>
>
>
>
> **Q2: If I didn't miss it, the article doesn't employ the L0 norm. If that's the case, I recommend not introducing L0 in the notation section. Additionally, in Equation 6 and subsequent formulas, if you are using the L2 norm, I suggest explicitly writing it as $||\cdot||_2$ not $||\cdot||$.**
>
> **A2:** Thanks. We have improved the writing accordingly in the revision.
>
> **Q3: Under section 3, "In this section, for the convenience of presenting our method DDPM......", I think it should be EDDPM.**
>
> **A3:** Thanks. We have corrected it in the revision.
>
> **Q4: Regarding the mask variable, m\_t, I understand it to be the t-th entry of the vector m. Following tradition, a single random variable should not be represented in bold form, and it is recommended to write as m\_t.**
>
> **A4:** Thanks. We have modified these notations in the revision.
>
> **Q5: Under Eq.(9), $\tilde{\mathbf{u}}(\mathbf{x}_t, \mathbf{x}_0),\tilde{\beta}_t$ should be written as $\tilde{\mathbf{u}}_t(\mathbf{x}_t, \mathbf{x}_0,\mathbf{m})$, $\tilde{\beta}_t(\mathbf{m})$, because they depend on $\mathbf{m}$.**
>
> **A5:** Thanks. We have corrected these typos in the revision.
>
> **Q6: Under Eq.(9),  in the definition of $\tilde{\beta}_t$, it has an extra “t" in the lower right corner.**
>
> **A6:** Thanks. We have corrected it in the revision.
>
> **Q7: Due to the L1 constraint on $\mathbf{s}$, most entries in will be pushed to 0. But why are the other entries pushed towards 1? Are there situations where some are around 0.5? In such cases, how should step $t$ be handled? Is there any explanation?**
>
> **A7:** Actually, our constraint on $\mathbf{s}$ enforces most of $s_i$
> converge to either 0 or 1. The detailed explainations are given in **A3 of Reviewer MPYQ** . In Figure 3 of the revision, we present more results on the histogram of the values of $s_i$ on CIFAR-10.  In all our experiments, we observed that few  elements of $s_i$ do not converge to  0 and 1, which can hardly been seen in the histogram. The elements are always close to 0 or 1, introducing negligible randomness in our final sampled denoising steps, i.e., when fully trained our denoising trajectory is almost deterministic.
>
> **Q8: Under Eqn.(11), it should be $\nabla_{\mathbf{\theta}} \Phi(\theta, \mathbf{s})$ and $\nabla_{\mathbf{s}} \Phi(\theta, \mathbf{s})$.**
>
> **A8:** Thanks. We have corrected it in the revision.
>
>
> **Q9: Eqn.(12) is confusing because $\gamma_e$ is not used in the algorithm. Is this a typo? Is it the case that $K=\gamma_e$ * T, and $\gamma_e$ is expressed as in equation 12?**
>
> **A9:** A more detailed expression of $\gamma_e$ is given in Eqn.(12) in the revision.  $\gamma_e$ is used to define the constraint region in the current epoch $e$ and we use the projected gradient descent to update $\mathbf{s}$ and gradient descent to train $\theta$. That is,
>
>   $ \theta = \theta - \eta \nabla_{\theta}\Phi (\theta, \mathbf{s}) \mbox{ and } \mathbf{s} = proj_{\mathcal{S}}\left(\mathbf{s} - \eta \nabla_{\mathbf{s}} \Phi (\theta, \mathbf{s})\right),$
>
> where $\mathcal{S} = \\{ \mathbf{s}\in \mathbb{R}^T: ||\mathbf{s}||_1\leq K_e, \mathbf{s}\in [0,1]^T \\}$
>  with $K_e = \gamma_e T.$
>
> $proj_{\mathcal{S}}(\cdot)$  is the projection on $\mathcal{S}$. The projection can be efficiently computed with the details added in Theorem 1 in the appendix of the revision. How the constraint induces sparsity on $\mathbf{s}$ is explained in **A3 of Reviewer MPYQ**.
>
>
> **Q10: In the context of image synthesis, the baseline comparison is limited. There are many acceleration sampling algorithms proposed for image synthesis, such as DPM-solver (Lu et al., 2022). While these works are mentioned in the related work section, they are not compared in the experiments.**
>
> **A10:** Due to the space limitation, we add 4 latest  sampling acceleration methods in Table 1 for comparison. Our EDDPM has more significant superiority  when using fewer denoising steps, especially achieves 4.89 FID with 5 steps.

---

> > ### Author Response · Authors · 2023-11-16
> > **Response to Reviewer jbL3 [2]**
> >
> > **Q11: There are several issues with the citation format and some references are duplicated or inappropriately cited in arXiv format, even though they have been published.**
> >
> > **A11:** Thanks for your detailed review. 1) We have deleted the duplicated references of papers [1-3]. 2) We have reviewed references individually and replaced outdated citation formats.
> >
> > [1] Fan Bao, Chongxuan Li, Jiacheng Sun, Jun Zhu, and Bo Zhang. Estimating the optimal covariance with imperfect mean in diffusion probabilistic models.
> >
> > [2] Alexander Quinn Nichol and Prafulla Dhariwal. Improved denoising diffusion probabilistic models.
> >
> > [3] Kashif Rasul, Calvin Seward, Ingmar Schuster, and Roland Vollgraf. Autoregressive denoising diffusion models for multivariate probabilistic time series forecasting.

---

> > > ### Author Response · Authors · 2023-11-20
> > > **Invitation for a Discussion**
> > >
> > > Dear Reviewer jbL3,
> > >
> > > Thank you very much for taking the time to review our paper and for your constructive comments.
> > >
> > > We have made extensive clarifications and discussions as you indicated. We hope  we have effectively addressed your concerns.
> > >
> > > The key points in our rebuttal include:
> > >
> > > 1. We added four latest sampling acceleration methods in Table 1. Compared with them, our EDDPM method exhibits advanced performance.
> > >
> > > 2. We clarified your misunderstanding in Figure 1.
> > >
> > >
> > > 3. We improved the writing accordingly.
> > >
> > > We are eager to hear your valuable opinion on the efforts we have made during the rebuttal period. If you have any further questions that you would like us to address, we are more than willing to discuss them in detail.
> > >
> > >
> > > We eagerly await your feedback and look forward to engaging in a fruitful discussion.
> > >
> > > Authors

---

> ### Comment · Reviewer_jbL3 · 2023-11-22
> **Reply to the authors' rebuttal**
>
> Thanks for the authors' feedback. Most of my concerns are addressed, and I decide to increase the score to 6.

---

> > ### Author Response · Authors · 2023-11-22
> > **Thank you for your positive feedback!**
> >
> > Dear Reviewer jbL3,
> >
> > Really appreciate your time to read our responses carefully and your positive comments.
> >
> > Thank you for your dedicated efforts in helping us refine this paper.
> >
> > Authors

---

### Official Review · Reviewer_v4Co · 2023-10-31

**Soundness:** 3 good
**Presentation:** 3 good
**Contribution:** 3 good
**Rating:** 8
**Confidence:** 3

**Summary:**

This paper proposes a new skipping scheme, EDDPM, for denoising diffusion models. EDDPM uses parameterized probabilistic masking to decide whether to skip a diffusion time step for faster sampling speed. The proposed method probabilistically reparameterizes the discrete masking selection problem into a tractable continuous optimization problem. The experiment shows significant improvement in generation time (without losing much quality), and the learned masking scheme will reduce to deterministic masking.

**Strengths:**

1. The proposed EDDPM method is new. The idea of adding learnable probabilistic masking to the diffusion model is reasonable and seems to be new in the literature.
2. The derivation of the method seems to be solid.
3. The performance of EDDPM is good.

**Weaknesses:**

1. There are some typos and citation errors to be fixed. For example: two periods appear at the end of the first paragraph, [Bao et al., 2022a] and [Bao et al., 2022b] are duplicates, etc.

**Questions:**

1. Is the sparse masking result (like Fig. 2 (b)) pervasive across different datasets?
2. I am wondering how you deal with the $\ell$-1 norm constraint on $s$ during training. Is it through projection? Is the training stable with the stochastic gradient estimator in Eq. (11)?
3. Do you have plans to release the code for open access?

---

> ### Author Response · Authors · 2023-11-16
> **Response to Reviewer v4Co**
>
> Thanks for your constructive comments.
>
> **Q1: There are some typos and citation errors to be fixed. For example: two periods appear at the end of the first paragraph, [Bao et al., 2022a] and [Bao et al., 2022b] are duplicates, etc.**
>
> **A1:** Thanks for your detailed review. We have revised them in the new version.
>
>
>
> **Q2: Is the sparse masking result (like Fig. 2 (b)) pervasive across different datasets?**
>
> **A2:** Yes, the constraint on $\mathbf{s}$ enforces  most of the  mask scores converge to either 0 or 1. The reasons are discussed in **A3 of Reviewer MPYQ**. We also presented the results on CIFAR-10 dataset in the appendix. In all our experiments, we observed that few  elements of $s_i$ do not converge to  0 and 1, which can hardly been seen in the histogram. The elements are always close to 0 or 1, introducing negligible randomness in our final sampled denoising steps, i.e, when fully trained our denoising trajectory is almost deterministic.
>
>
>
> **Q3: I am wondering how you deal with the l-1 norm constraint on s during training. Is it through projection? Is the training stable with the stochastic gradient estimator in Eq. (11)?**
>
> **A3:** Yes, we adopt projected gradient descent to update $\mathbf{s}$ in each iteration. We would like to point out that the projection can be computed efficiently, whose details are given in Theorem 1 in the appendix of the revision. The training is stable since the dimension of $\mathbf{s}$ is relatively low compared with the model weights $\theta$.
>
> **Q4: Do you have plans to release the code for open access?**
>
> **A4:** Yes, we will release the code soon.

---

### Official Review · Reviewer_1NMC · 2023-10-31

**Soundness:** 3 good
**Presentation:** 3 good
**Contribution:** 3 good
**Rating:** 8
**Confidence:** 2

**Summary:**

This paper proposes a new method called EDDPM to improve the sampling efficiency of denoising diffusion models. The key idea is to assign a probabilistic binary mask to each diffusion timestep, indicating whether it should be skipped or kept. The mask probabilities are learned jointly with the diffusion model to identify and eliminate redundant steps.  EDDPM is evaluated on image synthesis and time series imputation tasks. It efficiently compresses diffusion models to 20% of steps yet achieves equal or better performance than baseline methods.

**Strengths:**

- The proposed work addresses the critical challenge of inefficient sampling in diffusion models via a very clever and novel probabilistic masking approach. The formulation and inference are elegant and impressive.

- The experiment settings and results are solid. It's impressive to see the proposed work can achieve state-of-the-art performance in time series imputation and image synthesis with 5-20% of steps, and enables efficient step-compression of large diffusion models through one-epoch.

The presentation is also clear and easy to follow.

**Weaknesses:**

- The policy-gradient-based update for the prob masking is more like reinforcement learning, rather than the Bayesian variational inference. As the classical VI-based update is also feasible for inferring the Bern distribution, more discussion is encouraged on why adopting the policy-gradient-based update.



- For the constrain $K$ of the total step, can the "Gradually Increasing Masking Rate" trick guarantee theoretically that the final learned step is constrained by $K$? My understanding is it controls the prob $p$ of the Bern distribution and the final steps are based on the random samples. Also, it's not very clear to me why the learned $s$ is doomed to be almost 0 or 1 . The L1 constraint can do it for sure. However, it seems the training procedure doesn't include the L1 constraint explicitly, but uses the  "Gradually Increasing Masking Rate" trick. Clarification on these points is encouraged.



- There are some typos and missing things that should be fixed. For example:
1. The statement under equation 11,  "we can estimate the gradients of $\nabla_{\mathbf{s}} \Phi(\theta, \theta)$"- should it be $\nabla_{\mathbf{s}} \Phi(\theta, s)$
2. The equation 12, what's the meaning of $e_1$?

**Questions:**

See weakness

---

> ### Author Response · Authors · 2023-11-16
> **Response to Reviewer 1NMC**
>
> Thanks for your constructive comments.
>
> **Q1: The policy-gradient-based update for the prob masking is more like reinforcement learning, rather than the Bayesian variational inference. As the classical VI-based update is also feasible for inferring the Bern distribution, more discussion is encouraged on why adopting the policy-gradient-based update.**
>
> **A1:** Our considerations in choosing the gradient estimators are as follows.
>
> (1) Bayesian method could not be a perfect choice for our training problem. The reason is that its performance is sensitive to the selection of priors. In the case of the Bernoulli distribution, the Beta distribution serves as the conjugate prior, displaying significant variation for different parameters, $\alpha$ and $\beta$. In the early stage, an inappropriate prior could make the model difficult to converge.
>
> (2) In contrast, for policy gradient, we design a schedule that gradually increase the masking ratio and we can update the model by projected gradient descent efficiently.
>
> Thanks for your suggestion and we will conduct an in-depth exploration on the combination of Bayesian methods and our framework in the future.
>
>
>
>
> **Q2: For the constrain K of the total step, can the "Gradually Increasing Masking Rate" trick guarantee theoretically that the final learned step is constrained by $K$? My understanding is it controls the prob $p$ of the Bern distribution and the final steps are based on the random samples. Also, it's not very clear to me why the learned $s$ is doomed to be almost 0 or 1 . The L1 constraint can do it for sure. However, it seems the training procedure doesn't include the L1 constraint explicitly, but uses the "Gradually Increasing Masking Rate" trick. Clarification on these points is encouraged.**
>
> **A2:** As most elements in  $\mathbf{s}$ converge to either 0 or 1, the length of the final learned denoising procedure is well constrained by K. The explanation of how the constraint induces sparsity on $\mathbf{s}$ is provided in **A3 of Reviewer MPYQ**.
>
> We impose the constrains on the training process as follows. For each epoch, we  define the following constraint region
> $$\mathcal{S} = \\{ \mathbf{s}\in \mathbb{R}^T: ||\mathbf{s}||_1\leq K_e, \mathbf{s}\in [0,1]^T \\},$$
> where $K_e = \gamma_e T$ with $\gamma_e$ defined in Eqn.(12). This allows us to control the sparsity of the mask $\mathbf{m}$ by managing the sum of the probabilities $s_i$. In each iteration, we adopt the projected  gradient descent to update $\mathbf{s}$ based on the constraint region above. The details are given in Eqn.(13) in the revision. As demonstrated by Theorem 1 in the appendix of the revision, this projection can be computed efficiently. As training progresses, $K_e$ gradually decreases to the targeted $K= \gamma_f T$, leading to a reduction in the number of sampling steps involved.
>
>
>
>
> **Q3: There are some typos and missing things that should be fixed. The equation 12, what's the meaning of $e_1$?**
>
> **A3:** We have provided more detailed expression for $\gamma_e$ in Eqn.(12). $e_1$ is a positive integer indicating that we train the entire denoising steps in the first $e_1$ epochs.

---

> > ### Comment · Reviewer_1NMC · 2023-11-20
> >
> > Thanks for the detailed response. It resolves my concerns and questions. I will hold the score and still support the paper.

---

> > > ### Author Response · Authors · 2023-11-20
> > > **Thank you for your positive feedback!**
> > >
> > > Dear Reviewer 1NMC,
> > >
> > > Really appreciate your time to read our responses carefully and for your positive comments.
> > >
> > > Authors

---

### Official Review · Reviewer_MPYQ · 2023-11-02

**Soundness:** 3 good
**Presentation:** 2 fair
**Contribution:** 3 good
**Rating:** 6
**Confidence:** 3

**Summary:**

This paper proposed a novel accelerated diffusion model, called Efficient Denoising Diffusion (EDDPM). EDDPM eliminates the need for manually selecting denoising steps in previous sampling acceleration methods via probabilistic masking. The probabilistic masking is parameterized to be a Bernoulli random variable and thus can be efficiently learned jointly with the model parameters. After full training, most of the probabilistic masks converge to deterministic values of either 0 or 1, retaining only a small number of important steps. Empirical results demonstrated the sampling efficiency of EDDPM over state-of-the-art sampling acceleration methods on two tasks including time series imputation and image generation.

**Strengths:**

- This paper is well-structured.
- The probabilistic masking technique is interesting and novel to the best of my knowledge.
- Empirically, EDDPM is more sample efficient than other baselines.

**Weaknesses:**

- The paper writing is not good for some parts.
- The authors did not align their work correctly in the literature.
- Some parts of the method are not clear.

**Questions:**

- The motivation part of this paper is not persuasive. The authors said that prior works often involve manual selection or the use of handcrafted rules, such as uniform skipping, to determine denoising steps. It is not entirely true for all efficient sampling methods. The authors are referred to check this survey [r1]. The authors may need to narrow down the scope of comparison.
  - [r1] Yang, Ling, Zhilong Zhang, Yang Song, Shenda Hong, Runsheng Xu, Yue Zhao, Wentao Zhang, Bin Cui, and Ming-Hsuan Yang. "Diffusion models: A comprehensive survey of methods and applications." ACM Computing Surveys (2022).
- The literature review part of “Acceleration of DPMs” is not well-written. The authors fail to position their work in the literature and summarize the issues of prior work because their chosen scope is too wide.
- Why is it true? “Due to the constraints on s, i.e., $\lVert s \rVert_1 \leq K$ and $s \in [0, 1]^T$, the optimal $s$ would be sparse and most of its components would be either 0 or 1.”
- Section 4.2: Should the masking rate decrease gradually instead? In Eqn. (12), $y_e$ also decreases as e increases from 1 to N. The smaller K is, the fewer the number of steps is. In addition, what is $e_1$?
- Algorithm 1.
  - What is the initialization of $s$?
  - It is unclear how to use $y_e$.
  - It is also unclear how the sparse constraint is enforced during training.
- Baselines. How is DDPM with 10% or 20% denoising steps implemented?
- Table 1. Although the algorithm is unclear, should we still expect EDDPM to become DDPM when using all the denoising steps?

**Minors**:
- Some citations are weird.
  - “However, this decoupled approach can lead to suboptimal performance (Song et al., 2020; Bao et al., 2022c; Liu et al., 2022; Bao et al., 2022b).” → It is unclear what is the purpose of putting citations here.
  - “Sohl-Dickstein et al. (Sohl-Dickstein et al., 2015) firstly introduced diffusion probabilistic models (DPMs) that they can convert one distribution into a target distribution, in which each diffusion step is tractable.” and many more→ There is a rule to put citations at the beginning of a sentence. Please follow it.
- “The training of diffusion models involves a weighted variational bound derived from the connection between diffusion probabilistic models and denoising score matching with Langevin dynamics..” This sentence is not correct.
- “Bao et al. (Bao et al., 2022c;b) proposed to estimate the optimal covariance in each timestep of the reverse process”. This sentence has a loose connection with previous sentences. It is unclear what are the benefits of the development.
- “(Luhman & Luhman, 2021) compressed the diffusion process by combining the GANs and DPMs, and the proposed model only needs one sampling step for generation.” This paper uses knowledge distillation. There is no combination of GANs and DPMs.
- The word reduced variance variational bound is confusing as reduced variance refers to another concept.
- There are some typos and grammatical errors, please correct it. To name a few:
  - our method DDPM → our method EDDPM
  - thorough → through
  - In Section 4.1: $\tilde{\beta}t = \frac{1 - \alpha{t-1}(m)}{1 - \alpha_t(m)} \beta_t m_t$
  - Line 10 in Algorithm 1: $\nabla_\theta \Phi(\theta, s) \to \nabla_s \Phi(\theta, s)$
- Eqn. (4) (and its related sentences) should be put above Eqn. (2).
- Bulleted listings should be avoided when writing. In Section 5, some parts bulleted listing should be converted to paragraphs.
- The caption of Table 1 lacks the notation of the second-best method.

---

> ### Author Response · Authors · 2023-11-16
> **Response to Reviewer MPYQ [1]**
>
> Thanks for your constructive comments.
>
> **Q1: The motivation part of this paper is not persuasive. The authors said that prior works often involve manual selection or the use of handcrafted rules, such as uniform skipping, to determine denoising steps. It is not entirely true for all efficient sampling methods. The authors are referred to check this survey [r1]. The authors may need to narrow down the scope of comparison.**
>
> **A1:** When referring to learning-free efficient sampling methods, it is common for them to rely on handcrafted rules. We appreciate your reminder. In our revision, we have adhered to the survey [r1], which categorizes existing studies into learning-free and learning-based methods. Specifically, we have focused on reviewing the most relevant approaches, particularly the learning-based ones, in the related work section.
>
> It is worth noting that learning-free methods often employ handcrafted rules, whereas learning-based methods decouple the training and inference schedules. This decoupling allows for separate learning of the training and sampling schedules. However, both approaches have the potential to result in suboptimal performance.
>
> **Q2: The literature review part of “Acceleration of DPMs” is not well-written. The authors fail to position their work in the literature and summarize the issues of prior work because their chosen scope is too wide.**
>
> **A2:** We appreciate your valuable suggestion. Given the extensive range of efficient sampling studies available, it is challenging to review all of them in the related work section. To ensure a comprehensive overview, we adhere to the survey [r1], which categorizes existing studies into learning-free and learning-based methods. In the revised version, we position our work within the realm of learning-based efficient sampling methods and focus on reviewing the most relevant studies in the related work section.
>
>
> **Q3: Why is it true? “Due to the constraints on $\mathbf{s}$, i.e., $\|\mathbf{s}\|_1 \leq K$ and $\mathbf{s}\in [0,1]^T$, the optimal would be sparse and most of its components would be either 0 or 1.”**
>
> **A3:** It is true and we would like to explain this property in the following three aspects:
>
> 1) Prof. Robert Tibshirani, the author of the well-known sparse learning method lasso, provides an explanation of this property from a geometric perspective in pages 10-12 (Fig.2.2) of his book titled **Statistical Learning with Sparsity: The Lasso and Generalizations**. To be precise, the optimization problem of lasso is equivalent to the following one with some $t$:
> $$\min_{\mathbf{\beta}}|| \mathbf{y}- \mathbf{X}\mathbf{\beta}||^2, \mbox{ s.t. } \sum_{i=1}^p |\beta_i| \leq t. $$
> Note that the constraint region above is a diamond ($p=2$) or a rhomboid ($p>2$). The optimal solution is the point, where the elliptical contours of the loss hit this constraint region. When the dimension $p=2$, “the diamond has corners; if the solution occurs at a corner, then it has one parameter $\beta_j$ equal to 0. When $p>2$, the diamond becomes a rhomboid, and has many corners, flat edges, and faces; there are many more opportunities for the estimated parameters to be zero" (refer to page 12 of the above book). The situation in our problem is essentially the same with lasso, the only difference is  that our constraint region $\sum_{t=1}^T |s_t| \leq K, \mathbf{s}\in [0,1]^T$ has more corners (i.e., the coordinates are 0 or 1) than that of lasso, therefore, the optimal $s_t$ has a high probability to be either 0 or 1.
>
> 2) We can also understand this property from the Complementary slackness in KKT condition, which says “$=$" has to be hold in the “$\leq$" inequality constraints ($s_t \leq 1$ and $-s_t \leq 0$ in our problem) has to be hold as long as the corresponding dual variables  are nonzero. The dual variables have a high probability to be nonzero, as they are not imposed with sparse constraints. Please refer to wiki page "https://en.wikipedia.org/wiki/Karush%E2%80%93Kuhn%E2%80%93Tucker_conditions". for more details.
>
> 3) We empirically verified this property in Fig.2(b).
>
>
> **Q4: Section 4.2: Should the masking rate decrease gradually instead? In Eqn.(12),
>  $\gamma_e$ also decreases as $e$ increases from $e_1$ to N. The smaller K is, the fewer the number of steps is. In addition, what is $e_1$?**
>
> **A4:** 1) $\gamma_e$ is the ratio of the remaining steps in the current epoch $e$. We have provided more detailed expression in Eqn.(12). 2) We train the full denoising sequence in the first $e_1$ epochs and then the remaining ratio $\gamma_e$ decreases gradually to the targeted ratio $\gamma_f$ according to Eqn.(12).

---

> > ### Author Response · Authors · 2023-11-16
> > **Response to Reviewer MPYQ [2]**
> >
> > **Q5: What is the initialization of s? It is unclear how to use $\gamma_e$. It is also unclear how the sparse constraint is enforced during training.**
> >
> > **A5:** The initialization of $\mathbf{s}$ is set to $\mathbf{1}\in \mathbb{R}^T$. $\gamma_e$ is used to define the constraint region in the current epoch $e$, given by
> >
> > $\mathcal{S} = \\{ \mathbf{s}\in \mathbb{R}^T: ||\mathbf{s}||_1\leq K_e, \mathbf{s}\in [0,1]^T \\},$
> >
> > where $K_e = \gamma_e T$. This allows us to control the sparsity of the mask $\mathbf{m}$ by managing the sum of the probabilities $s_i$. We adopt the projection operator to ensure that the sum of $s_i$ is always no larger than $K_e$ during epoch $e$. As training progresses, $K_e$ gradually decreases to the targeted $K= \gamma_f T$, leading to a reduction in the number of sampling steps involved. Specifically, we update $\mathbf{s}$ using projected gradient descent and $\theta$ using regular gradient descent, as follows:
> >
> > $\theta = \theta - \eta \nabla_{\theta}\Phi (\theta, \mathbf{s})$  and $\mathbf{s} = proj_{\mathcal{S}}\left(\mathbf{s} - \eta \nabla_{\mathbf{s}} \Phi (\theta, \mathbf{s})\right),$
> >
> > where operator $proj_{\mathcal{S}}(\cdot)$, i.e., the projection on $\mathcal{S}$ can be efficiently computed. More details on the projection can be found in Theorem 1 in the appendix. The explanation of how the constraint induces sparsity on the optimal solution $\mathbf{s}$ is provided in A3.
> >
> >
> >
> >
> > **Q6: How is DDPM with 10\% or 20\% denoising steps implemented?**
> >
> > **A6:** It is implemented through uniform skipping, as shown in Table 2, it can always achieve better performance than quadratic skipping.
> >
> >
> > **Q7: Although the algorithm is unclear, should we still expect EDDPM to become DDPM when using all the denoising steps?**
> >
> > **A7:** Yes. When all sampling steps are used, they are exactly the same, which is consistent with our results in Table 1.
> >
> >
> > **Q8: “However, this decoupled approach can lead to suboptimal performance (Song et al., 2020...).” → It is unclear what is the purpose of putting citations here. “Sohl-Dickstein et al. (Sohl-Dickstein et al., 2015) firstly introduced diffusion probabilistic models (DPMs) that ...” and many more→ There is a rule to put citations at the beginning of a sentence. Please follow it.**
> >
> > **A8:** 1) Thanks for your reminder. In the revision, we followed the survey [r1] and reorganized our introduction, during which this issue was addressed. 2) We eliminated this issue by replacing the citation format $\citep$ with $\citet$ .
> >
> >
> > **Q9: “The training of diffusion models involves a weighted variational bound derived from the connection between diffusion probabilistic models and denoising score matching with Langevin dynamics.” This sentence is not correct.**
> >
> > **A9:** Thanks. We have deleted this statement in the revision.
> >
> > **Q10: “Bao et al. (Bao et al., 2022c;b) proposed to estimate the optimal covariance in each timestep of the reverse process”. This sentence has a loose connection with previous sentences. It is unclear what are the benefits of the development.
> > “(Luhman \& Luhman, 2021) compressed the diffusion process by combining the GANs and DPMs, and the proposed model only needs one sampling step for generation.” This paper uses knowledge distillation. There is no combination of GANs and DPMs.**
> >
> > **A10:** 1)We have improved the related work section in the revision and addressed this issue.
> > 2) Thanks for detailed review. Our citation was incorrect and we have revised it in the revision.
> >
> >
> > **Q11: The word reduced variance variational bound is confusing as reduced variance refers to another concept.**
> >
> > **A11:** Initially, we followed the terminology presented in Appendix A of the DDPM paper. We have now replaced it with the term “variational lower bound”.
> >
> >
> > **Q12: There are some typos and grammatical errors, please correct it. Eqn. (4) (and its related sentences) should be put above Eqn. (2). Bulleted listings should be avoided when writing. In Section 5, some parts bulleted listing should be converted to paragraphs. The caption of Table 1 lacks the notation of the second-best method.**
> >
> > **A12:** Thanks. We have corrected them accordingly in the revision.

---

> > > ### Author Response · Authors · 2023-11-20
> > > **Invitation for a Discussion**
> > >
> > > Dear Reviewer MPYQ:
> > >
> > > Thank you very much for taking the time to review our paper and for your constructive comments.
> > >
> > > We have made extensive clarifications and discussions as you indicated. We hope we have effectively addressed your concerns.
> > >
> > > The key points in our rebuttal include:
> > >
> > > 1. We gave the detailed reasons why the elements of our score vector $\mathbf{s}$ converge to either 0 or 1. A similar case has been presented in the textbook "Statistical Learning with Sparsity: The Lasso and Generalizations" written by the leading scientist, Prof. Robert Tibshirani, i.e., the author of lasso. Please refer to the details in **A3** in our author response to you.
> > >
> > > 2. We improved the related work section based on survey [r1], and emphasized the differences between our proposed EDDPM and previous learning-based sampling methods.
> > >
> > >
> > > 3. We further gave more detailed expressions for the  details of method in Eqn.(12) and Eqn.(13). We provided  the reason why the projection in Eqn.(13) can be efficiently computed in Theorem 1 in the appendix.
> > >
> > > We are eager to hear your valuable opinion on the efforts we have made during the rebuttal period. If you have any further questions that you would like us to address, we are more than willing to discuss them in detail.
> > >
> > >
> > > We eagerly await your feedback and look forward to engaging in a fruitful discussion.
> > >
> > >
> > > Authors

---

> ### Comment · Reviewer_MPYQ · 2023-11-20
> **Respone to authors' rebuttal**
>
> I thank the authors for patiently reading and responding to all my questions. I found that my concern was adequately addressed. In addition, the revised paper has improved in terms of both the related works and methodology parts.
>
> Further note: The authors should add A3 to the appendix to ease the understanding of the paper.
>
> I would like to increase my score to 6 and incline to an acceptance.

---

> > ### Author Response · Authors · 2023-11-21
> > **Thank you for your positive feedback!**
> >
> > Dear Reviewer MPYQ,
> >
> > Really appreciate your time to read our responses carefully and for your positive comments.  If you have any additional suggestions or comments, please don't hesitate to share them. We are always happy to answer your questions and engage in further discussion.
> >
> > Authors

---

> ### Author Response · Authors · 2023-11-21
> **A3 is included in the new version.**
>
> Dear Reviewer MPYQ,
>
> According to your suggestion, we have included A3 into Section C of the appendix and updated the submission. Thanks.
>
> Authors

---

### Author Response · Authors · 2023-11-16
**Response to ACs and all the reviewers**

Thanks to the ACs and  reviewers for your great efforts and time. We thank all reviewers for your valuable comments. In the process of discussions with you, we have answered the questions and revised the paper according to the reviewers' initial suggestions. To be precise,

1) We gave a detailed explaination on why adding our constraint to $\mathbf{s}$ can make most of  its elements converge to either 0 or 1.

2) As suggested by Reviewer MPYQ, we have improved the related work section in the revision.

3) According to the concerns of Eqn.(12) and how we use our contraint region during training, we have provided more detailed expression in the reversion. We also provided the detailed steps for updating the parameters $\mathbf{s}$ and $\theta$ in Eqn.(13).

4) We  fixed all typos and incorrect citations mentioned by reviewers one by one.


Last but not least, we would like to thank all reviewers again for their valuable comments to make our work more solid and thorough. If there are any further suggestions or comments, please feel free to raise them and we will always be happy to answer your questions and resolve your concerns. In closing, thank you to all the reviewers.

---

### Author Response · Authors · 2023-11-20
**General Response and Invitation for a Discussion**

Dear ACs and all reviewers:


We sincerely appreciate the time and efforts of the reviewers in providing their valuable feedback. We have incorporated the suggested modifications in our manuscript, which are highlighted in red.

The key points of our rebuttal can be summarized as follows:

1. We gave the detailed reasons why the elements of our score vector $\mathbf{s}$ converge to either 0 or 1. A similar case has been presented in the textbook "Statistical Learning with Sparsity: The Lasso and Generalizations" written by the leading scientist of AI, Prof. Robert Tibshirani, i.e., the author of lasso.

2. We improved the related work section based on survey [r1], and emphasized the differences between our proposed EDDPM and previous learning-based sampling methods.

3. We added four latest sampling acceleration methods in Table 1. Compared with them, our EDDPM method exhibits advanced performance.


4. We further gave more detailed expressions for the  details of method in Eqn.(12) and Eqn.(13). We provied  the reason why the projection in Eqn.(13) can be efficiently computed in Theorem 1 in the appendix.

We eagerly await feedback from the reviewers and look forward to engaging in a fruitful discussion.

Authors

---

### Author Response · Authors · 2023-11-21
**Explanation on the sparisty of the score has been inclued in the revision.**

Dear Reviewers,

According to the suggestion of Reviewer MPYQ in the new feedback, we have included the explanation on the sparsity of the score vector, i.e., A3 of the authors response to Reviewer MPYQ, in Section C of the appendix and updated the submission.

 If there are any further suggestions or comments, please feel free to raise them and we will always be happy to answer your questions and resolve your concerns.

Authors